# NOT ALL MODELS SUIT EXPERT OFFLOADING: ON LOCAL ROUTING CONSISTENCY OF MIXTURE-OF-EXPERT MODELS

**Jingcong Liang**
Fudan University
jcliang22@m.fudan.edu.cn

**Siyuan Wang**[*]
University of Southern California
sw_641@usc.edu

**Miren Tian & Yitong Li & Duyu Tang**
Huawei Technologies Ltd.
tianmiren1,liyitong3,tangduyu@huawei.com

**Zhongyu Wei**[*]
Fudan University &
Shanghai Innovation Institute
zywei@fudan.edu.cn

## ABSTRACT

Mixture-of-Experts (MoE) enables efficient scaling of large language models (LLMs) with sparsely activated experts during inference. To effectively deploy large MoE models on memory-constrained devices, many systems introduce *expert offloading* which caches a subset of experts in fast memory, leaving others on slow memory to run on CPU or load on demand. While some research has exploited the locality of expert activations, where consecutive tokens activate similar experts, the degree of this **local routing consistency** varies across models and remains understudied. In this paper, we propose two metrics to measure local routing consistency of MoE models: (1) **Segment Routing Best Performance (SRP)**, which evaluates how well a fixed group of experts can cover the needs of a segment of tokens, and (2) **Segment Cache Best Hit Rate (SCH)**, which measures the hit rate of an expert cache utilizing a length of future information under a cache limit. We analyze 20 MoE LLMs with diverse sizes and architectures and use toy models to verify key factors related to local routing consistency. We find a strong trade-off between local routing consistency and *local* load balance, while showing that *global* load balance can coexist with local routing consistency. Meanwhile, settings like shared experts that decrease expert combination space can lead to low local routing consistency. We further reveal that domain-specialized experts contribute more to routing consistency than vocabulary-specialized ones, and that most models balance between cache effectiveness and efficiency with cache sizes approximately twice the active experts. These findings pave the way for memory-efficient MoE design and deployment without compromising inference speed. We publish the code for replicating experiments at https://github.com/ljcleo/moe-lrc.

## 1 INTRODUCTION

Mixture-of-Experts (MoE) is a widely adopted model architecture in many large language models (LLMs) that enables efficient model size scaling through sparse activation(Fedus et al., 2022; Jiang et al., 2024; Dai et al., 2024; Abdin et al., 2024). MoE models replace dense feed-forward networks (FFNs) with multiple expert modules, with only a subset activated during inference. However, the vanilla implementation requires all experts to be loaded into memory, restricting its application on memory-constrained devices such as mobile phones. To address this limitation, the *expert offloading* technique has been proposed to allow partial loading of expert modules during inference (Eliseev & Mazur, 2023; Hwang et al., 2024; Yi et al., 2025). Specifically, expert offloading caches a subset of experts in fast memory (e.g., GPU memory) based on predefined heuristics, while storing remaining experts in slower but larger-capacity storage (e.g., CPU memory or disk). During inference, especially

---

[*]Corresponding authors.

in the decoding stage, if a token activates an expert not cached in fast memory, the system either computes the expert forward results with CPU and slow memory (Kamahori et al., 2024; Tang et al., 2024), or unloads a cached expert according to specific rules such as Least Recently Used (LRU), and replaces it with the demanded expert (Kong et al., 2024; Zhong et al., 2025).

However, frequent CPU offloads or on-demand loading within a short period can significantly degrade the efficiency of the expert offloading system and slow down inference, particularly when processing lengthy contexts with inevitable topic shifts. Prior research has focused on optimizing the design of the expert offloading system, aiming to strategically select which experts to cache in a given context to maximize cache hit rates. Among them, some observed and exploited the locality of expert activations, where similar routing choices appear within a consecutive segment of tokens, thereby minimizing the need for CPU offloads and on-demand loading (Eliseev & Mazur, 2023; Xue et al., 2024b; Zhang et al., 2025). This is especially beneficial during the decoding phase, where tokens are generated one after another.

Nevertheless, not all MoE models exhibit such continuous routing patterns uniformly, and the degree or frequency of this phenomenon varies across models. Understanding this variance may help design MoE architectures that are friendly to expert offloading systems and vice versa. In this work, we investigate the degree of this inherent consecutive routing property, which we term **local routing consistency**, of different MoE-based LLMs to explore their potential effectiveness in segment-based expert routing or caching. Figure 1 illustrates how different levels of local routing consistency reflect different routing patterns. Specifically, we propose two metrics that quantitatively reflect the local routing consistency of a specific model. (1) **Segment Routing Best Performance (SRP)** measures how effectively a segment router that selects a fixed group of experts for all tokens in a segment can approximate the original router's decisions. SRP not only reflects local routing consistency *without parameters other than segment length*, but also enables analyzing activation patterns of *individual experts*. (2) **Segment Cache Best Hit Rate (SCH)** represents the hit rate of an oracle expert cache that evicts unused experts based on the activation frequency within a specific length of future, under a cache size limit related to the number of active experts. SCH measures local routing consistency on model- and router-levels, yet is more related to the performance of *real expert offloading systems* as it accounts for the true cache limit.

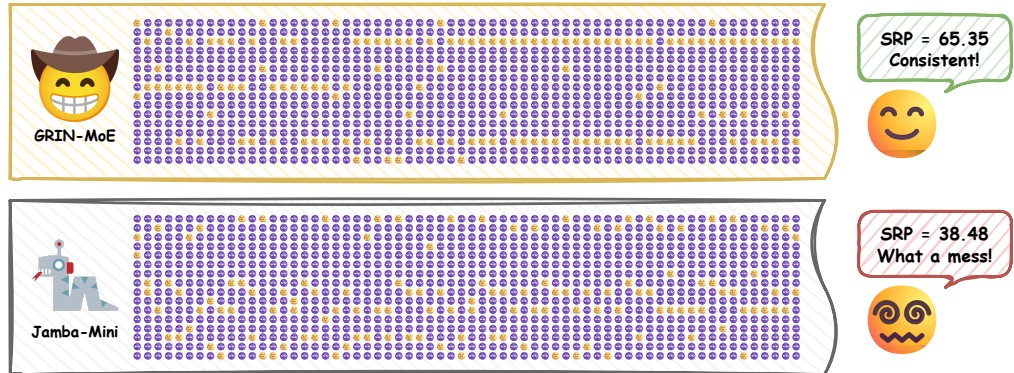

Figure 1: Routing results by GRIN-MoE (Liu et al., 2024) layer 21 and Jamba-Mini-1.6 (Lenz et al., 2025) layer 25 on the same input (Java code). Despite having similar model sizes and the same number of experts, GRIN-MoE exhibits more consistent routing patterns than Jamba-Mini-1.6, activating certain experts continuously, so expert caching with it will be more feasible and effective.

We conduct experiments on 20 MoE-based LLMs, covering parameter scales ranging from 3 billion to 54 billion and encompassing diverse architectures. While most models exhibit similar local routing consistency within a few tokens, the variance enlarges when the segment length increases. Further verification on toy models reveals a **trade-off relation between local routing consistency and *local* load balance**; however models can **achieve good *global* load balance with high local routing consistency**. Meanwhile, **Shared experts harms local routing consistency**, as well as expert configurations that **limit expert combination space**. Additionally, we investigate local routing consistency across different context domains and assess its relationship with experts' domain and vocabulary preferences. The findings reveal that **domain-specialized experts**, if they exist,

**contribute more** to local routing consistency, whereas vocabulary specialization has less impact. Finally, we verify the strong correlation between SCH and hit rates of common cache algorithms, confirming the bond between local routing consistency and expert offloading efficiency. We conclude that most MoE models can achieve **optimal balance** between segment caching effectiveness and deployment efficiency with a **cache size approximately 2x the number of active experts**.[*]

Overall, our contributions are threefold:

1. We propose *local routing consistency*, a property of MoE models that reflects the potential efficiency of expert offloading for the model. We design two metrics to quantify local routing consistency: segment routing best performance (SRP), which provides parameter-free fine-grained analysis, and segment cache best hit rate (SCH), aligned with practical expert offloading.
2. We conduct empirical analysis across 20 MoE-based LLMs, identifying and verifying (with toy models) key factors that affect local routing consistency, such as local load balance, shared experts, and expert combination space. We also reveal that domain-specialized experts contribute more to local routing consistency than vocabulary-specialized ones, as well as global load balance.
3. We analyze SCH under different cache sizes relative to the number of active experts. Alongside the optimal cache size derived from SRP results, we conclude that cache sizes 2x the size of active parameters achieve the best segment caching results on most models.

## 2 DEFINITIONS

### 2.1 PRELIMINARY: MIXTURE OF EXPERTS

Transformer-based language models have most parameters on feed-forward network (FFN) layers. As the model size scales up, LLMs often replace them with sparse MoE layers to reduce computational cost during inference. A typical MoE layer with $E$ experts can be parameterized by $E$ smaller FFNs $F_1(; \theta_1), \ldots, F_E(; \theta_E)$, where each $F_i \colon \mathbb{R}^d \to \mathbb{R}^d$ defines a single expert. There is also a router $R \colon \mathbb{R}^d \to \mathbb{R}^E$ in the MoE layer to choose experts and give weights. For each token $x$ with hidden representation $h_x \in \mathbb{R}^d$, its output is given by

$$[s_1, \ldots, s_E] = \mathrm{Softmax}(R(h_x)); \; [w_1, \ldots, w_E] = \mathrm{Top}_k(s_1, \ldots, s_E); \; o_x = \sum_{i=1}^{E} w_i F_i(h_x; \theta_i) \quad (1)$$

where $\mathrm{Top}_k$ preserves the $k$ largest scores and sets others to $0$. Experts with a null score can be effectively deactivated without calculating, thus saving computational resources. Other components in the transformer architecture may also be replaced by their MoE variant, such as Mixture-of-Attention (Zhang et al., 2022) for self-attention and MixLoRA (Li et al., 2024a) for LoRA adapters. Nevertheless, we focus on MoE layers that replace FFNs, as this is the most prominent and effective design (in terms of the number of parameters). More discussions about MoE LLMs can be found in Appendix B, where we also briefly review expert offloading for MoE models.

The above routing procedure is done token by token, which does not guarantee consecutive routing. Since the consistency of consecutive routing decisions can benefit expert offloading systems, it is necessary to investigate the degree of consecutive routing of different MoE-based LLMs. In the following sections, we propose two metrics to measure this **local routing consistency** and compare them across MoE models with different structure parameters.

### 2.2 SEGMENT ROUTING BEST PERFORMANCE (SRP)

An intuitive way to measure local routing consistency is to compare the distribution of routing choices between tokens in a continuous segment. However, typical metrics for distribution comparison, such as the Kullback-Leibler divergence, are designed for two distributions and are therefore unsuitable for our purpose. Instead, we propose **Segment Routing best Performance (SRP)** (Figure 2), which measures how well a simplified, segment-based router can mimic the behavior of the original token-based router. Here we give a brief introduction of SRP, with formal definition in Appendix C.1.

---

[*]We propose this insight from the aspect of model design, especially when building models for devices with known memory constraints (e.g., mobile phones).

Figure 2: An illustration of segment routing best performance (SRP). Left: SRP on a single expert, where a segment estimator gives segmented predictions on every segment, and SRP is the best possible $F_1$ score. Right: SRP on a group of experts, where a segment router predicts which experts are activated in a segmented manner, and SRP is the upper bound of $F_1$.

**On a single expert**   Given an input sequence, the activation pattern of an expert in an MoE model on this sequence can be seen as a series of binary classification tasks. A segment estimator with segment length $m$ aims to predict this activation sequence, but in a segmented manner: At position $i$, it predicts the same result (active or not) for positions from $i$ to $i + m - 1$. We select the $F_1$ score to measure the performance of the estimator, because missing major activations is worse than activating on minor activations (Kong et al., 2024; Skliar et al., 2025); recall (hit-rate) is also inappropriate as we do not set up an upper bound on how frequently the expert can be activated. Moreover, to diminish positional effects, we consider all possible positions and treat each position as an individual binary classification task with $m$ samples. Based on all the settings above, the SRP on this expert is defined as the upper bound $F_1$ of such estimators.

Despite modelling using a virtual segment estimator, SRP itself solely depends on the expert and the segment length: The $F_1$ score is maximized if and only if the estimator gives active predictions for all segments that had the expert activated more times than a specific threshold; a formal proof is given in Appendix C.3. Therefore, SRP is an intrinsic property of the expert that reflects its local routing consistency, unrelated to any specific routing methods.

**On a group of experts**   For a group of experts (in the same layer or in the same model), we can combine single-expert SRPs for *expert-choice* routing (Zhou et al., 2022), since experts are independent of each other. However, in the more traditional and prevalent *token-choice* routing, experts are not activated independently of one another. Therefore, we use a segment router that decides which experts should be activated, also in a segmented manner. Similar to the single expert case, we define SRP as the upper bound of $F_1$ between the routing decisions of the original router and the segment router, respectively. For a group of experts, SRP measures how well the original router coordinates the experts to achieve layer-level or model-level routing consistency.

Like SRP on a single expert, we do not limit the number of activated experts the segment router can choose at each position. Nevertheless, we do not want it to activate too many experts to achieve a high $F_1$. Therefore, we define the **segment routing size ratio** $\hat{\rho}$ as the ratio between the average number of activated experts of the segment router and the original router (which is usually a fixed number). A small ratio indicates that the local routing consistency of the experts is high enough, so that segment routing does not need to select too many experts to cover real demands under average cases. We use it as a supplementary metric to distinguish between cases where groups of experts have similar segment routing best performances.

### 2.3   SEGMENT CACHE BEST HIT RATE (SCH)

The advantage of SRP as a local routing consistency metric is that it solely relies on the expert (group) and the segment length, making it suitable for analyzing individual experts. However, real expert offloading scenarios usually have a hard cache size limit, so the best global $F_1$ score may not be achievable. Moreover, when considering caching performance, $F_1$ score is not as straightforward as hit rate (recall). Therefore, we propose **Segment Cache best Hit rate (SCH)** (Figure 3), which is more related to expert offloading.

Following the expert group case, instead of a segment router with unlimited activation settings, we now consider a segment cache with a hard cache size limit. More specifically, the cache has a specific

cache ratio $\rho$, which is the ratio between the cache size and the number of activated experts. This cache works like the other caches in expert offloading systems: for each token, it loads (or preloads) the demanded experts not in the cache while evicting the same number of unused experts.[†] The difference is that it evicts experts that are activated the least times in the next $m$ tokens. Under this setting, SCH is defined as the hit rate of the segment cache. Due to their similar segment-based behavior, SCH can act as a bridge between SRP and the efficiency of real expert offloading systems.

Note that the upper bound hit-rate of any cache is given by the clairvoyant replacement algorithm that evicts the expert whose next activation will occur farthest in the future. Although SCH is also modeled on a cache algorithm that relies on some oracle information, the required information is easier to learn and predict (future activation frequency vs. precise time of next activation). Thus, practical cache algorithms are more likely to reach a cache hit rate close to SCH. Furthermore, in Section 5.2 we show that SCH is already close to the optimal cache hit rate under certain conditions. Therefore, we stick to SCH to align with SRP-based results.

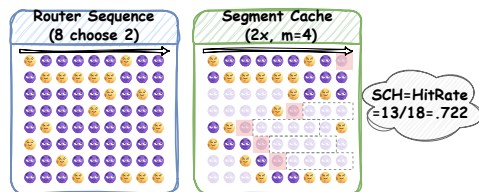

Figure 3: An illustration of segment cache best hit rate (SCH). An oracle expert cache evicts experts (red shade) that are least activated in the next $m$ tokens (grey dash box); SCH is its hit rate.

# 3 SRP-BASED CONSISTENCY ANALYSIS

## 3.1 EXPERIMENT SETUP

**Models**   We first conduct experiments on 20 MoE-based LLMs with model sizes ranging from 3B to 57B, covering both popular (SwitchTransformers, Mixtral) and recent (DeepSeek-V2, Qwen3) models. We list their architecture and configuration details in Appendix D.1. We may also use shorter names (e.g., OLMoE and Qwen3) when there is no ambiguity. Given that many models also have post-trained (e.g., SFT, RL) versions, we compare the local routing consistency between base and post-trained versions of several models in Appendix E.1, where we find no significant difference. Therefore, we stick to the base version in our main experiments.

To validate our observations on existing MoE models concerning architecture design, we further pretrain a series of OLMoE-like toy models from scratch, and conduct the same observational experiments on them. The baseline model has 1.43B parameters, with shrunk depth and hidden dimensions compared to the original OLMoE. Each other model modifies one key architectural or training parameter, such as expert granularity, number of shared experts, and load balance loss coefficient, while maintaining the same total number of parameters (1.43B). Appendix D.2 provides more details about the model and training configurations. In the following sections, we refer to the above two groups of models as REAL and TOY, respectively.

**Dataset**   We construct our sample corpus $S$ from two sources: (1) **Generic Corpora:** We include all 7 categories from RedPajama (Together Computer, 2023), including C4, CommonCrawl, Books, Wikipedia, ArXiv, StackExchange, and GitHub. (2) **Downstream Application:** We append several datasets with cases aligned with modern LLM applications, including arena-human-preference-140k (LMArena; LMArena, 2025, OpenMathInstruct-2 (OpenMath; Toshniwal et al., 2025), OpenCode-Instruct (OpenCode; Ahmad et al., 2025), and OpenScienceReasoning-2 (OpenScience; NVIDIA Corporation, 2025). We treat each RedPajama category and downstream application dataset as a distinct domain, and refer to its subset corpus using the data source (Books, GitHub, etc.). The full corpus contains 22,528 input samples, each sample having 512 tokens. Appendix D.3 gives more details on the data processing and input generation process.

**Method and configuration**   We collect every MoE layer's routing decisions for every input[‡], and for each expert, count the number of activated tokens $f$ in every segment. Although modern LLMs utilize position encodings to distinguish tokens from different positions, we demonstrate in Appendix E.2

---

[†]We only describe the decoding stage; prefilling is similar except that one expert may handle multiple tokens.
[‡]In encoder-decoder models, encoder layers only consider encoder input, and decoder layers likewise.

that segments from different positions have nearly identical SRP, except for the very first ones that may contain special head tokens. Therefore, we perform calculations on all segments and do not care about their positions. To obtain SRP, we count the number of segments with the same $f$ for each expert (group), then compute the $F_1$ score for every $\alpha$ candidate, choose the $\alpha$ that achieves the highest $F_1$ and finally obtain the segment routing best performance and size ratio.

## 3.2 OVERALL RESULTS

Figure 4 illustrates the SRP of each REAL model under various segment lengths $m$. While most models have similar SRP and $\hat{\rho}$ when $m = 4$, the difference between models becomes significant as the segment length increases, yet the relative order remains after $m = 16$. There is a gap between short-term ($m = 4$) and long-term ($m \geq 16$) local routing consistency, where **many models exhibit the short-term one but only a few demonstrate the long-term one**.

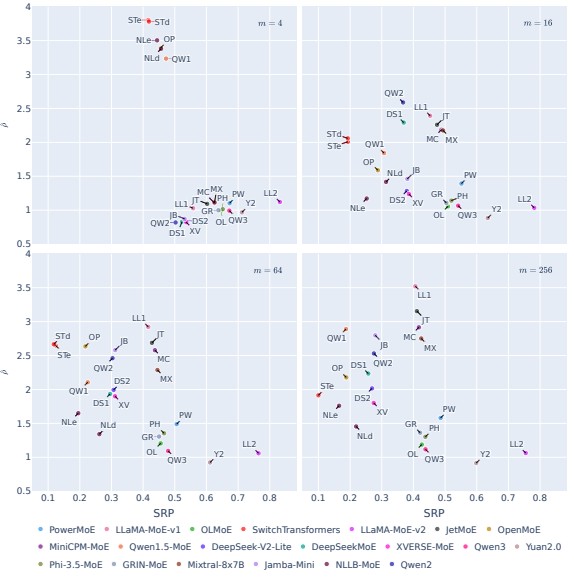

Figure 4: SRP and $\hat{\rho}$ of REAL models on the full corpus.

Figure 5: SRP and log perplexity of TOY models on the full corpus. We only show $m = 16$ results for simplicity.

We roughly divide REAL into four groups that have similar SRP characteristics, whose SRP when $m = 16$ are demonstrated in Table 1:

- Group 1 (LLaMA-MoE-v2–OLMoE) has the highest SRP ($> 0.5$ when $m = 16$) and $\hat{\rho}$ ($\sim 1.25$) across all segment lengths, showing strong long-term local routing consistency.
- Group 2 (Mixtral-8x7B–LLaMA-MoE-v1) have the second highest SRP ($\sim 0.48$ when $m = 16$), but their long-term $\hat{\rho}$ becomes high ($\sim 2.5$).
- Group 3 (XVERSE-MoE–DeepSeekMoE) has a significantly lower SRP than group 2, especially the long-term one ($\sim 0.36$ when $m = 16$), but their $\hat{\rho}$ is a bit lower $\sim 2.0$.
- Group 4 (NLLB-MoE–SwitchTransformers) have the lowest SRP ($< 0.31$ when $m = 16$); contrasting other models, their short-term $\hat{\rho}$ is already high, but their long-term $\hat{\rho}$ becomes lower.

## 3.3 WHAT AFFECTS LOCAL ROUTING CONSISTENCY?

Table 1 already provides clues about the relationship between local routing consistency (SRP) and model architecture. Nevertheless, due to the vast heterogeneity among REAL models, we further validate possible factors with TOY models, results illustrated in Figure 5. Significant points include:

**Load balance** This is one of the most intuitive factors related to local routing consistency, which is another key feature for efficient MoE inference (Lepikhin et al., 2021). For instance, both DeepSeek-AI et al. (2024b) and Skliar et al. (2025) suggest adding bias to router outputs. Still, the former promotes *little activated* experts for load balance and the latter promotes *recently cached* experts for

Table 1: REAL model SRP in descending order compared with several architecture parameters. "A:T": ratio between active and all experts; "S:A": ratio between shared and active experts; "every $x$": apply MoE every $x$ layers; "after 1st": apply MoE after the first layer.

| Model | SRP | MoE | A:T | S:A | Model | SRP | MoE | A:T | S:A |
|---|---|---|---|---|---|---|---|---|---|
| LLaMA-MoE-v2 | 78.16 | all | 1:4 | 0 | XVERSE-MoE | 38.58 | all | 3:32 | 1:3 |
| Yuan2.0 | 63.48 | all | 1:16 | 0 | Jamba-Mini | 38.08 | every 2 | 1:8 | 0 |
| PowerMoE | 55.17 | all | 1:5 | 0 | DeepSeek-V2-Lite | 37.92 | after 1st | 3:32 | 1:3 |
| Qwen3 | 54.14 | all | 1:16 | 0 | DeepSeekMoE | 36.94 | after 1st | 3:32 | 1:3 |
| Phi-3.5-MoE | 51.98 | all | 1:8 | 0 | Qwen2 | 36.74 | all | 1:8 | 1:1 |
| OLMoE | 50.91 | all | 1:8 | 0 | | | | | |
| GRIN-MoE | 50.39 | all | 1:8 | 0 | NLLB-MoE (encoder) | 25.24 | every 4 | 1:64 | 0 |
| | | | | | (decoder) | 31.35 | | | |
| Mixtral-8x7B | 49.36 | all | 1:4 | 0 | Qwen1.5-MoE | 30.71 | all | 1:15 | 1:1 |
| MiniCPM-MoE | 48.85 | all | 1:4 | 0 | OpenMoE | 28.77 | every 6 | 1:16 | 1:2 |
| JetMoE | 47.45 | all | 1:4 | 0 | SwitchTF (encoder) | 19.33 | every 2 | 1:128 | 0 |
| LLaMA-MoE-v1 | 45.29 | all | 1:4 | 0 | (decoder) | 19.27 | | | |

effective caching. To investigate their relation, we compute the standard deviation (SD) of all experts' activation frequencies in a model and compare it with SRP in Table 2. Many REAL models with high SRP exhibit imbalanced routing, which largely contributes to their local routing consistency (see Appendix E.7). We also verify this on TOY models, where NoLB reaches a high SRP but has very poor load balance, while OverLB, which further prioritizes load balance during training, has very low SRP. Although load balance is also important for efficient training, from the aspect of expert offloading applications (e.g., edge computing), it is still worth **trading some load balance for local routing consistency if expert offloading will be involved.**

Table 2: SRP ($m = 16$) and load balance (LB) measured by activation frequency SD of experts.

| Model | SRP | LB | Model | SRP | LB | Model | SRP | LB |
|---|---|---|---|---|---|---|---|---|
| LLaMA-MoE-v2 | 78.16 | 29.04 | XVERSE-MoE | 38.58 | 2.71 | NoLB | 56.42 | 13.21 |
| Yuan2.0 | 63.48 | 13.86 | Jamba-Mini | 38.08 | 3.05 | ActMore | 55.69 | 6.54 |
| PowerMoE | 55.17 | 12.90 | DeepSeek-V2-Lite | 37.92 | 2.34 | DenseFst | 44.87 | 4.25 |
| Qwen3 | 54.14 | 3.19 | DeepSeekMoE | 36.94 | 2.03 | DenseHlf | 43.67 | 3.71 |
| Phi-3.5-MoE | 51.98 | 4.90 | Qwen2 | 36.74 | 6.74 | Baseline | 43.56 | 4.02 |
| OLMoE | 50.91 | 6.79 | NLLB-MoE (en) | 25.24 | 1.75 | FewerExp | 41.62 | 3.63 |
| GRIN-MoE | 50.39 | 3.89 | (de) | 31.35 | 2.13 | 1ShrExp | 41.38 | 3.43 |
| Mixtral-8x7B | 49.36 | 2.70 | Qwen1.5-MoE | 30.71 | 0.58 | 2ShrExp | 38.79 | 3.06 |
| MiniCPM-MoE | 48.85 | 2.59 | OpenMoE | 28.77 | 2.56 | OverLB | 36.42 | 1.79 |
| JetMoE | 47.45 | 1.12 | SwitchTF (en) | 19.33 | 0.58 | ActFewer | 27.13 | 1.14 |
| LLaMA-MoE-v1 | 45.29 | 2.66 | (de) | 19.27 | 0.66 | | | |

Note that models like Qwen3 and GRIN-MoE show high SRP and moderate load balance simultaneously. Since high local routing consistency almost always means low *local* load balance, we conclude that these models exhibit good *global* load balance: While a single query may only activate a portion of experts, queries from different topics are likely to activate different sets of experts, eventually covering all experts. Section 4 further reveals that these models possess **strong domain-specialized experts** that help increase **both local routing consistency and global load balance**.

**Shared experts and expert combination space** Besides training objectives, model architecture can also play an important role in forming local routing consistency, the most significant of which we found is the existence of shared experts: Among REAL models, all models in groups 1 and 2 do not use shared models; Among TOY models, Share1 and Share2, having similar perplexity levels with Baseline, show significantly lower SRP than Baseline. We suggest two potential reasons for why **shared experts harm local routing consistency:** One reason could be the bypass effect, where more information is processed by shared experts, making the real MoE part less important. Another reason, which is linked to multiple MoE design factors, is the decreased size of expert combination space, which is also mentioned by Muennighoff et al. (2025). More specifically, the existence of shared experts decreases the number of both available and activated experts, resulting in fewer possible expert

combinations for routing. This may prevent the router from making local adjustments on routing decisions between consecutive tokens, resulting in low local routing consistency. In fact, among the TOY models, 32Exp (use fewer experts) and Top2 (activate fewer experts) do exhibit lower SRP than Baseline, while Top16 (activate more experts but less than half) show higher SRP. This further demonstrates that **more expert combinations benefit local routing consistency**. Nevertheless, this is a less significant factor compared to load balance and shared experts, as some REAL models (e.g., Phi and GRIN) do not strictly follow this rule.

**Interleaved MoE layers**   Different from introducing dense components (shared experts) *inside* MoE layers, *interleaving or concatenating* MoE layers with dense ones seems to have less significant impact. Although Skip1 (first layer dense) and Sparse2 (MoE every 2 layers) possess higher SRP than Baseline, the lead is minor, and both models suffer from high perplexity. Meanwhile, all REAL models that involve dense layers fall into groups 3 and 4. However, the low SRP of these models may not be due to dense layers: they either use shared experts (e.g., DeepSeekMoE) or activate experts sparsely (e.g., SwitchTransformers), both of which can contribute to low local routing consistency.

## 4   LOCAL ROUTING CONSISTENCY AND EXPERT SPECIALIZATION

### 4.1   DOMAIN-WISE LOCAL ROUTING CONSISTENCY

In Section 3, we analyze models on the full corpus, which consists of text data from 11 different domains. However, each domain has its token distributions, which may affect the router decision's distribution. Figure 6 illustrates the relative difference between domain-wise and global SRP of each model when $m = 16$, where we observe three different patterns among all models: (1) Models like Phi-3.5-MoE, GRIN-MoE, and OLMoE have significantly **higher SRP on ArXiv, StackExchange and GitHub**, whose SRP can be more than 10% higher than global SRP. They also exhibit higher SRP on OpenMath and OpenCode, indicating specialized experts for math and coding tasks. (2) Models like LLaMA-MoE-v2, Yuan2.0, and Qwen3 have significant **higher SRP on Wikipedia and other generic domains** but lower on OpenMath and OpenCode. They seem to have specialized experts for generic text (e.g., multilingual experts) instead of math or coding. (3) Models like Mixtral-8x7B, MiniCPM-MoE and JetMoE have **similar SRP across all domains** with insignificant differences. All of them have mediocre to low SRP. Interestingly, we found that all TOY models show the first pattern (like the original OLMoE) regardless of architectural or training configuration tweaks, indicating that **the formation of such patterns may stem from the pretraining data distribution.**

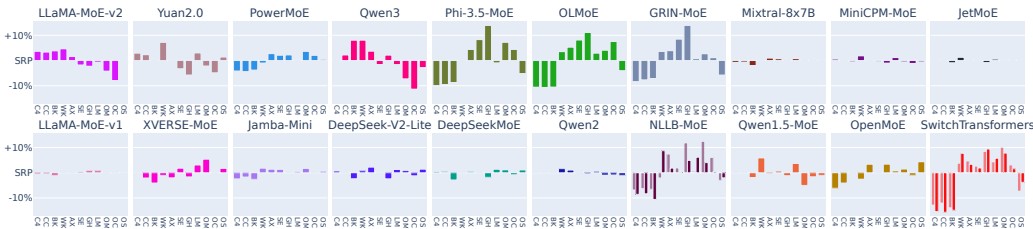

Figure 6:  SRP ($m = 16$) on each domain corpus, relative to SRP on the full corpus. In-group bars (from left to right): C4, CommonCrawl, Books, Wikipedia, ArXiv, StackExchange, GitHub, LMArena, OpenMath, OpenCode, and OpenScience. For encoder-decoder models, light colors represent the encoder and dark colors represent the decoder.

### 4.2   EXPERT SPECIALIZATION

We argue that domain-wise local routing consistency patterns appear across models due to specialized experts in each model. To clarify this, we consider two types of expert specialization, first introduced by Muennighoff et al. (2025): (1) **Domain specialization:** the normalized frequency of an expert being activated on tokens from a specific domain. We compute the coefficient of variation (CV) of activation frequency across all domains as a domain-free metric. (2) **Vocabulary specialization:** the normalized frequency of an expert being activated on a specific token ID. We follow Muennighoff

et al. (2025) to obtain the vocabulary specialization of each expert. We compare each model's SRP, average expert specialization, and the correlation between its experts' specialization and SRP in Figure 7; expert distribution between specialization and SRP is also demonstrated in Appendix E.7.

**Domain specialization**    Many REAL models show a positive correlation between their experts' domain specialization and SRP. Exceptions include LLaMA-MoE-v2, which constantly activates a group of experts, resulting in very high SRP; Qwen2 and LLaMA-MoE, meanwhile, hardly have any domain specialization. In contrast, Qwen3, Phi-3.5-MoE, GRIN-MoE, and OLMoE exhibit high SRP, high average expert domain specialization, and strong correlation between them simultaneously. As mentioned in Section 3.3, these models also demonstrate global load balance (see Table 2); with domain specialized experts this is explanable: A domain-specialized expert tends to be active when the context comes from certain domains or topics, but not others. Therefore, when the context matches, the expert is likely to be consistently activated (local routing consistency), but as long as the context becomes unrelated, the expert becomes inactive (global load balance).

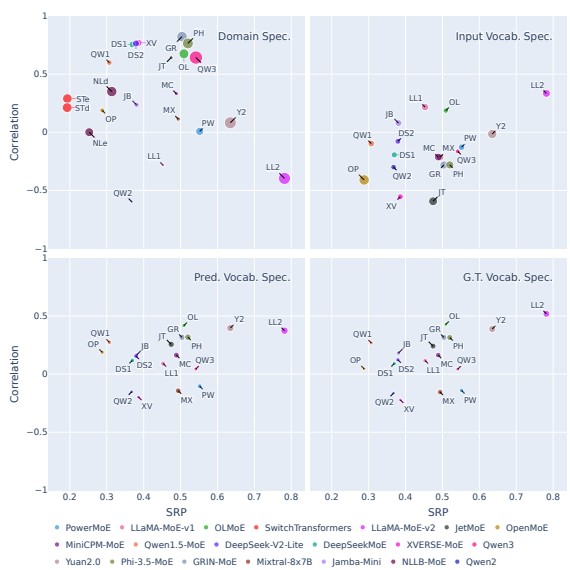

Figure 7: SRP, expert specialization, and their correlation. Correlation between SRP and expert specialization is *per model*, based on single-expert results.

**Vocabulary specialization**    We consider three kinds of vocabulary specialization on the input, the model's predicted output, and the ground-truth, respectively (Muennighoff et al., 2025). Most models demonstrate negative or insignificant correlation between *input* vocabulary specialization and SRP, except for LLaMA-MoE-v2 due to its constantly activated experts. On the other hand, SRP is slightly positively correlated to *prediction* or *ground* truth vocabulary specialization. We conjecture that such specialization happens more in later layers (Muennighoff et al., 2025) that process high-level information related to the context topic.

Above all, we can see that **domain specialization plays a more important role in local routing consistency than vocabulary specialization**, especially on models with both high local routing consistency and global load balance.

## 5    SCH-BASED CONSISTENCY ANALYSIS

### 5.1    OVERALL RESULTS

As mentioned in Section 2.3, SRP has several flaws that hinder its application in expert offloading. This section focuses on the segment cache best hit rate (SCH), which works with a size limit, to obtain a more straightforward insight into expert offloading and cache management. We calculate each model and each layer's SCH on every possible cache size by simulating the oracle cache described in Section 2.3 and recording the hit rate.

Figure 8 illustrates SCH of REAL models under different $m$s and $\rho$s. We can easily identify the four groups of models mentioned in Section 3.2 starting from $m = 16$: Group 1 models have the fastest growing SCH with respect to $rho$ when $\rho$ is small, as well as turning points near $\rho = 2$, after which they share similar SCH with group 2 models. Meanwhile, models from groups 3 and 4 have relatively low SCH, growing nearly linearly as $\rho$ increases. Since only group 1 models (with the highest local routing consistency) have turning points on SCH, we claim that in general, $\rho = $ **2 can balance cache effectiveness and efficiency.**

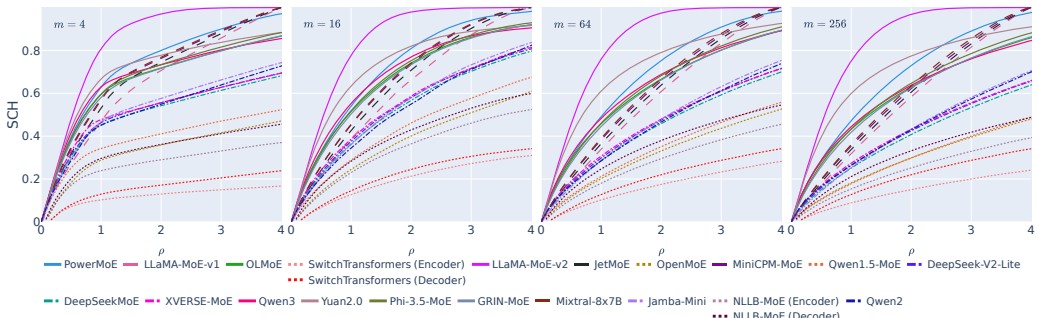

Figure 8: SCH of REAL models on the full corpus under different segment length $m$ and cache ratio $\rho$. Solid line: group 1; dashed: group 2; dash-dotted: group 3; dotted: group 4.

## 5.2 SCH AND COMMON CACHE ALGORITHM

Since SCH is based on an ideal cache system that relies on oracle information, it is crucial to understand how well it is correlated with real-world implementations. Table 3 lists the correlation between SCH and the cache hit rate of two widely adopted cache algorithms: least recently used (LRU) and least frequently used (LFU). All compared cache algorithms have hit rates highly correlated to SCH, even for short segments ($m = 4$). Furthermore, Appendix E.4 reveals that SRP and SCH are also highly correlated with each other. This suggests that models with higher local routing consistency tend to achieve higher expert cache hit rates, hence greater performance gain with expert offloading.

Furthermore, we use the optimal cache hit rate given by the clairvoyant replacement algorithm as a baseline, and compared the relative hit rate of the mentioned cache algorithms in Figure 4 (using Baseline as an example). When the cache size is moderate, SCH ($m = 16$) is very close to the optimal hit rate, while LRU and LFU consistently have hit rates lower than SCH. From this point, SCH can act as a "practical" ideal case for analytical analysis on expert offloading.

Table 3: Correlation between SCH and hit rate of cache algorithms across REAL models. LRU: least recently used; LFU: least frequently used.

| $m$ | LRU | LFU | Fixed |
|---|---|---|---|
| 4 | 81.20 | 77.39 | 76.26 |
| 16 | 90.43 | 88.70 | 89.79 |
| 64 | 93.10 | 92.82 | 95.50 |
| 256 | 97.52 | 99.20 | 97.91 |

Table 4: Baseline's SCH ($m = 16$) and hit rates of LRU and LFU relative to the optimal hit rate under different cache ratios $\rho$. The optimal hit rate is normalized to 100.

| $\rho$ | LRU | LFU | SCH |
|---|---|---|---|
| 1.0 | 56.49 | 61.92 | 80.97 |
| 2.0 | 67.04 | 70.87 | 90.55 |
| 3.0 | 75.26 | 78.35 | 96.23 |

## 6 CONCLUSION

In this paper, we investigate the property of MoE LLMs where similar experts can be continuously activated, namely *local routing consistency*. We propose two metrics to measure this property: segment routing best performance (SRP) and segment cache best hit rate (SCH). We compare SRP and SCH between multiple models and identify several key designs that may help improve local routing consistency of MoE LLMs, including local load balance trade-off, (no) shared experts and (enlarging) expert combination space. We also identify that domain-specialized experts contribute more to local routing consistency and help achieving global load balance. We further suggest that a cache size around 2x the number of active experts can balance cache effectiveness and efficiency.

**Acknowledgements** The work is supported by AI for Science Program, Shanghai Municipal Commission of Economy and Informatization (2025-GZL-RGZN-BTBX-02028). The project's computational resources are partially supported by CFFF platform of Fudan University.

**Ethics statement**    Our analytic methods and results still help design new MoE LLMs friendly to expert offloading and enable deployment on resource-constrained edge devices. While our study will likely have an indirect social impact, developers implementing local routing consistency to build more powerful LLMs must take responsibility for their products' societal implications.

**Reproducibility statement**    We constructed our sample corpus with deterministic algorithms, and we will publish the sampled corpus to ensure reproducibility. We also conducted all experiments with a deterministic configuration and will release relevant source code.

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

## A   USE OF LLMS

We primarily employ LLMs to polish the writing of this paper, utilizing tools such as Grammarly and Writeful. In all other cases, LLMs are the object of our experiments and analyses, and we do not use them for other purposes.

## B   RELATED WORK

### B.1   MoE-BASED LLM AND EXPERT ANALYSIS

Since its introduction into large neural networks, MoE has become a critical strategy to build large language models up to trillions of parameters (Shazeer et al., 2017; Fedus et al., 2022; Rajbhandari et al., 2022). While some early models like SwitchTransformers (Fedus et al., 2022) and NLLB (NLLB Team et al., 2022) employ encoder-decoder structures as their backbone, due to the success of GPT-3, the most recent popular MoE-based LLMs use decoder-only structures (Jiang et al., 2024; Yang et al., 2024a; DeepSeek-AI et al., 2024b; Abdin et al., 2024), replacing their original FFN layers with MoE layers containing multiple experts (other components may be replaced too, e.g., self-attention (Shen et al., 2023; 2024b) and LoRA (Li et al., 2024a; Feng et al., 2024)). Cai et al. (2024) systematically introduces MoE architectures and implementation in LLMs.

The popularity of MoE LLMs has triggered interest in understanding how experts are activated in such models. Many model reports and individual studies focused on the relation between expert selection and the input context. For example, Muennighoff et al. (2025) reported that OLMoE shows a significant difference in expert activity across different domains. Contrastively, Xue et al. (2024a) found that the routing choice of OpenMoE is highly related to the input token rather than the input context. Other works investigated the similarity among expert activation patterns (Li et al., 2024b; Lu et al., 2024), as well as the relation between expert output and routing choice (Pham et al., 2024; Lo et al., 2025). Some further proposed methods to reinforce such patterns (Guo et al., 2025; Chen et al., 2025). However, few of them have focused on the local activation pattern of experts. For example, Jiang et al. (2024) reported that in Mixtral, experts are more likely to be activated consecutively, compared to the random case. While their results provide fundamental support for many efficient MoE inference systems Liu et al. (2026), they only examined the case of 2 consecutive tokens, which may be insufficient to ensure the consistency of expert activation in longer segments.

### B.2   EFFICIENT MoE INFERENCE AND EXPERT OFFLOADING

The discrete nature of routers and redundant parameters has caused MoE models to infer more slowly and consume more memory than dense models with the same number of activated parameters. Many techniques have been proposed to boost the inference of MoE models, ranging from model modifications like model compression (Chen et al., 2022; Huang et al., 2025; Yang et al., 2024b; Rajbhandari et al., 2022) and soft routing (Muqeeth et al., 2024; Zhong et al., 2024) to system implementations like load-balanced expert parallel (Lepikhin et al., 2021; Huang et al., 2023; Li et al., 2023) and hardware adaptation (DeepSeek-AI et al., 2024b; Yi et al., 2025). Liu et al. (2026) provides an in-depth summary of various inference optimization strategies of MoE models.

In this paper, we mainly focus on the potential performance of expert offloading, which enables lossless inference of MoE models on memory-constrained devices by caching only some experts on (fast) memory while leaving others on slow memory or disk storage. Many such systems use pretrained external models and/or information from previous layers to prefetch experts for upcoming layers (Ren et al., 2024; Du et al., 2024; He et al., 2024; Song et al., 2024). Various expert offloading systems propose curated heuristics to manage the expert cache (Skliar et al., 2025; Xue et al., 2024b; Yu et al., 2025; Fang et al., 2025). Among them, some examine the locality of expert activations as empirical support for expert caching efficiency:

- Jiang et al. (2024) first observed that Mixtral-8x7B is likely to choose the same expert for the next token, with probabilities higher than the random expectation. Eliseev & Mazur (2023) extended the argument to 2-4 consecutive tokens through a case study, and boosted inference performance of the same model with LRU caching (plus other techniques such as prefetching and quantization). These

works align with our settings, but their analyses are limited to a single model (Mixtral-8x7B), short token spans, and lack a systematic and quantitative study.

- Xue et al. (2024b) reported frequent expert reuse during decoding, and observed that the reused experts depend on the prefilled input. Zhang et al. (2025) found similar routing choices between prefilling and decoding stages. Both observations are utilized to back the effectiveness of expert caching, yet are too coarse in terms of locality (at the input level instead of the token span level).

As more MoE LLMs emerge, understanding what models are more friendly to expert offloading becomes important for the development of both MoE architectures and expert offloading methods.

## C  FORMAL DEFINITIONS AND PROOFS

### C.1  FORMAL DEFINITION OF SRP

**Single expert**  For any input sequence $T = [t_1, \ldots, t_{|T|}]$, we denote the activation sequence of expert $e$ on $T$ as $A(e, T) = [a_1, \ldots, a_{|T|}]$, where $a_i \in \{0, 1\}$ indicates whether $e$ is activated on $t_i$ ($a_i = 1$) or not ($a_i = 0$). A segment-based router $R_e^m$ with segment length $m > 0$ for expert $e$ will try to mimic $[a_p, \ldots, a_{p+m-1}]$ for any $T$ and $p$. Let $R_e^m(T, p) = [b_1, \ldots, b_m]$ be its prediction, with the segment-prediction constraint:

$$b_i = 0, \ \forall i = 1, \ldots, m \quad \text{or} \quad b_i = 1, \ \forall i = 1, \ldots, m \tag{2}$$

In other words, $R_e^m$ decides that $e$ is either always active or always inactive on $[t_p, \ldots, t_{p+m-1}]$. For simplicity, we write $R_e^m(T, p) = 0$ and $R_e^m(T, p) = 1$ for the two cases respectively. By treating each segment routing attempt as a binary classification task with $m$ samples, and considering all possible segments of all possible inputs, we can calculate the $F_1$ score of $R_e^m$:

$$F_1(R_e^m) = \frac{2 \sum_T \sum_{p=1}^{|T|-m+1} R_e^m(T, p) \cdot f(e, T, p, m)}{\sum_T \sum_{p=1}^{|T|-m+1} [m \cdot R_e^m(T, p) + f(e, T, p, m)]} \tag{3}$$

where $f(e, T, p, m) = \sum_{i=p}^{p+m-1} A(e, T)[i]$ is the active frequency of $e$ in the segment of $T$ with length $m$ starting at position $p$. We demonstrate the detailed process to obtain this equation in Appendix C.2. Based on Equation 3, we define the segment routing best performance of $e$ under segment length $m$ as the maximum $F_1$ score any $R_e^m$ can achieve: $\text{SRP}(e, m) \triangleq \max_{R_e^m} F_1(R_e^m)$. Furthermore, in Appendix C.3 we prove that $F_1(R_e^m)$ is maximized if and only if $R_e^m$ gives active predictions for all segments that activates $e$ at least $\alpha_e^m$ times, where $\alpha_e^m \in [0, m]$ is only related to $e$ and $m$:

$$\text{SRP}(e, m) = \frac{2 \sum_T \sum_{f(e,T,p,m) \geq \alpha_e^m} f(e, T, p, m)}{\sum_T \sum_{p=1}^{|T|-m+1} [m \cdot I[f(e, T, p, m) \geq \alpha_e^m] + f(e, T, p, m)]} \tag{4}$$

Therefore, $\text{SRP}(e, m)$ is an intrinsic property of the expert that reflects its local routing consistency, unrelated to any specific segment routing methods.

**Expert group**  For a group of experts $E$, let $R_E^m$ be a segment-based router that decides whether each expert $e \in E$ should be activated in a segment of some input $T$ with length $m$; more specifically, $R_E^m(e, T, p)$ is a prediction sequence similar to $R_e^m(T, p)$ that also follows the segment-prediction constraint (Equation 2). Following the same procedure in Appendix C.2, we have

$$F_1(R_E^m) = \frac{2 \sum_T \sum_{p=1}^{|T|-m+1} \sum_{e \in E} R_E^m(e, T, p) \cdot f(e, T, p, m)}{\sum_T \sum_{p=1}^{|T|-m+1} \sum_{e \in E} [m \cdot R_E^m(e, T, p) + f(e, T, p, m)]} \tag{5}$$

Again, $F_1(R_E^m)$ is maximized if and only if $R_E^m$ gives active predictions for all expert-segment pairs where the expert is activated at least $\alpha_e^m$ times in the segment, where $\alpha_e^m$ is decided by $E$ and $m$. Therefore we have

$$\text{SRP}(E, m) \triangleq \max_{R_E^m} F_1(R_E^m) = \frac{2 \sum_T \sum_{f(e,T,p,m) \geq \alpha_e^m} f(e, T, p, m)}{\sum_T \sum_{p=1}^{|T|-m+1} \sum_{e \in E} [m \cdot I[f(e, T, p, m) \geq \alpha_e^m] + f(e, T, p, m)]} \tag{6}$$

$\text{SRP}(E, m)$ measures how well a group of experts is coordinated by the original router(s) to achieve layer-level or model-level local routing consistency.

## C.2 PROOF OF EQUATION 3

In Appendix C.1, we consider each routing decision of $R_e^m$ for a segment of length $m$ as a binary classification task with $m$ samples. If we merge all samples from all segments of all possible inputs into one global binary classification task, and define $f(e, T, p, m) = \sum_{i=p}^{p+m-1} A(e, T)[i]$ as in Appendix C.1, we will have the following prediction statistics:

$$
\begin{aligned}
\text{TP}(\text{R}_e^m) &= \sum_T \sum_{p=1}^{|T|-m+1} \sum_{i=1}^m I[A(e, T)[p + i - 1] = 1 \wedge R_e^m(T, p)[i] = 1] \\
&= \sum_T \sum_{p=1}^{|T|-m+1} R_e^m(T, p) \cdot f(e, T, p, m) \tag{7}
\end{aligned}
$$

$$
\begin{aligned}
\text{FP}(\text{R}_e^m) &= \sum_T \sum_{p=1}^{|T|-m+1} \sum_{i=1}^m I[A(e, T)[p + i - 1] = 0 \wedge R_e^m(T, p)[i] = 1] \\
&= \sum_T \sum_{p=1}^{|T|-m+1} R_e^m(T, p)[m - f(e, T, p, m)] \tag{8}
\end{aligned}
$$

$$
\begin{aligned}
\text{FN}(\text{R}_e^m) &= \sum_T \sum_{p=1}^{|T|-m+1} \sum_{i=1}^m I[A(e, T)[p + i - 1] = 1 \wedge R_e^m(T, p)[i] = 0] \\
&= \sum_T \sum_{p=1}^{|T|-m+1} [1 - R_e^m(T, p)] f(e, T, p, m) \tag{9}
\end{aligned}
$$

Therefore we have

$$
\begin{aligned}
F_1(R_e^m) &= \frac{1}{1/\text{Precision}(R_e^m) + 1/\text{Recall}(R_e^m)} \\
&= \frac{1}{[TP(R_e^m) + FP(R_e^m)]/TP(R_e^m) + [TP(R_e^m) + FN(R_e^m)]/TP(R_e^m)} \\
&= \frac{2TP(R_e^m)}{[TP(R_e^m) + FP(R_e^m)] + [TP(R_e^m) + FN(R_e^m)]} \\
&= \frac{2 \sum_T \sum_{p=1}^{|T|-m+1} R_e^m(T, p) \cdot f(e, T, p, m)}{\left[ \sum_T \sum_{p=1}^{|T|-m+1} m \cdot R_e^m(T, p) \right] + \left[ \sum_T \sum_{p=1}^{|T|-m+1} f(e, T, p, m) \right]} \\
&= \frac{2 \sum_T \sum_{p=1}^{|T|-m+1} R_e^m(T, p) \cdot f(e, T, p, m)}{\sum_T \sum_{p=1}^{|T|-m+1} [m \cdot R_e^m(T, p) + f(e, T, p, m)]}
\end{aligned} \tag{10}
$$

which gives Equation 3. $\square$

## C.3 PROOF OF EQUATION 4

Assume that $R_e^m(T_0, p_0) = 0$ for some $e$, $m > 0$, $R_e^m$, $T_0$ and $p_0$, then we have

$$
\begin{aligned}
F_1(R_e^m) &= \frac{2 \sum_T \sum_{p=1}^{|T|-m+1} R_e^m(T, p) \cdot f(e, T, p, m)}{\sum_T \sum_{p=1}^{|T|-m+1} [m \cdot R_e^m(T, p) + f(e, T, p, m)]} \\
&= \frac{2 \sum_T \sum_{p=1}^{|T|-m+1} R_e^m(T, p) \cdot f(e, T, p, m)}{m \sum_T \sum_{p=1}^{|T|-m+1} R_e^m(T, p) + \sum_T \sum_{p=1}^{|T|-m+1} f(e, T, p, m)} \\
&= \frac{2 \sum_{T \neq T_0 \wedge p \neq p_0} R_e^m(T, p) \cdot f(e, T, p, m)}{m \sum_{T \neq T_0 \wedge p \neq p_0} R_e^m(T, p) + \sum_T \sum_{p=1}^{|T|-m+1} f(e, T, p, m)} \\
&= \frac{2X}{mY + Z}
\end{aligned} \tag{11}
$$

where

$$X = \sum_{\substack{T \neq T_0 \\ p \neq p_0}} R_e^m(T, p) \cdot f(e, T, p, m), \quad Y = \sum_{\substack{T \neq T_0 \\ p \neq p_0}} R_e^m(T, p), \quad Z = \sum_T \sum_{p=1}^{|T|-m+1} f(e, T, p, m)$$

Let $\widehat{R_e^m}$ be a copy of $R_e^m$ except that $R_e^m(T, p) = 1$; all other routing decisions remain the same. Then the $F_1$ score of the new segment router will be

$$
\begin{aligned}
F_1\left(\widetilde{R_e^m}\right) &= \frac{2 \sum_T \sum_{p=1}^{|T|-m+1} \widetilde{R_e^m}(T, p) \cdot f(e, T, p, m)}{\sum_T \sum_{p=1}^{|T|-m+1} \left[ m \cdot \widetilde{R_e^m}(T, p) + f(e, T, p, m) \right]} \\
&= \frac{2[X + f(e, T_0, p_0, m)]}{m(Y + 1) + Z} \\
&= \frac{(mY + Z) \cdot F_1(R_e^m) + 2f(e, T_0, p_0, m)}{m(Y + 1) + Z} \\
&= \frac{(mY + Z) \cdot F_1(R_e^m) + m \cdot [2f(e, T_0, p_0, m)/m]}{(mY + Z) + m}
\end{aligned}
\tag{12}
$$

which is a weighted mean of $F_1(R_e^m)$ and $2f(e, T_0, p_0, m)/m$ with weights $mY + Z$ and $m$. Note that $Z = \sum_T \sum_{p=1}^{|T|-m+1} f(e, T, p, m) \geq 0$, and $Z = 0$ if and only if $f(e, T, p, m) = 0$ for all $T$ and $p$. If $Z = 0$, then $e$ is inactive everywhere and $F_1(R_e^m) = 0$ for any $R_e^m$[§], thus $\mathrm{SRP}(e, m) = 0$ and we can simply let $\alpha_e^m = 0$. Therefore, we assume that $Z > 0$, then both $m$ and $mY + Z$ are positive. Hence, $F_1\left(\widetilde{R_e^m}\right) \geq F_1(R_e^m)$ if and only if $2f(e, T_0, p_0, m)/m \geq F_1(R_e^m)$. Equality is achieved when and only when all equalities hold.

The above result indicates that, in order to increase $F_1(R_e^m)$, for any segment satisfying $R_e^m(T, p) = 0$ and $f(e, T, p, m) \geq (m/2) \cdot F_1(R_e^m)$, we should change the routing decision to $R_e^m(T, p) = 1$[¶], and for any segment satisfying $R_e^m(T, p) = 1$ and $f(e, T, p, m) < (m/2) \cdot F_1(R_e^m)$, we should change the routing decision to $R_e^m(T, p) = 0$. Under the case where the number of possible inputs is finite (which is the case for most LLMs due to their limited context windows), this will eventually result in a $\widehat{R_e^m}$ that activates and only activates all segments with $f(e, T, p, m) \geq (m/2) \cdot F_1\left(\widehat{R_e^m}\right)$, whose $F_1$ cannot increase further. Such $\widehat{R_e^m}$ must be unique and maximizing $F_1(R_e^m)$: Otherwise, if there exists another $\widehat{R_e^m}'$ with $F_1\left(\widehat{R_e^m}'\right) > F_1\left(\widehat{R_e^m}\right)$, then the only segments where $\widehat{R_e^m}$ and $\widehat{R_e^m}'$ disagree are the ones satisfying $(m/2) \cdot F_1\left(\widehat{R_e^m}\right) \leq f(e, T, p, m) < (m/2) \cdot F_1\left(\widehat{R_e^m}'\right)$, where $\widehat{R_e^m}(T, p) = 1$ and $\widehat{R_e^m}'(T, p) = 0$; however, changing $\widehat{R_e^m}$ on these segments to $0$ should not increase $F_1\left(\widehat{R_e^m}\right)$, thus $F_1\left(\widehat{R_e^m}'\right) \leq F_1\left(\widehat{R_e^m}\right)$, a contradiction. Therefore, we can let $\alpha_{e,m} = \left\lceil F_1\left(\widehat{R_e^m}\right) \right\rceil$, which yields Equation 4. □

## D EXPERIMENT SETUP DETAILS

### D.1 REAL MODEL ARCHITECTURE LIST

Table 5 lists the detailed architecture and configuration of all REAL models where we conduct our experiments.

A few notes:

- SwitchTransformers-Base-128 and NLLB-MoE-54B are encoder-decoder models that use the T5 architecture. SwitchTransformers-Base-128 has 12 encoder layers and 12 decoder layers. NLLB-MoE-54B has 24 encoder layers and 24 decoder layers.

---

[§]If $Y = R_e^m(T, p) = 0$ for all $T$ and $p$, then $F_1(R_e^m)$ is undefined, which we do not concern.
[¶]When $f(e, T, p, m) = (m/2) \cdot F_1(R_e^m)$, changing $R_e^m(T, p)$ does not affect $F_1(R_e^m)$.

Table 5: REAL Model architecture and configuration, sorted by model size. Experts: T: total; A: active; S: shared (not included in total).

| Model | # Params (B) | | # Layers | MoE Layer | # Experts | | |
| --- | --- | --- | --- | --- | --- | --- | --- |
| | Total | Active | | | T | A | S |
| PowerMoE-3B (Shen et al., 2024c) | 3.30 | 0.88 | 32 | all | 40 | 8 | 0 |
| LLaMA-MoE-v1-3.5B (Zhu et al., 2024) | 6.74 | 3.50 | 32 | all | 16 | 4 | 0 |
| OLMoE-1B-7B-0125 (Muennighoff et al., 2025) | 6.92 | 1.28 | 16 | all | 64 | 8 | 0 |
| SwitchTransformers-Base-128 (Fedus et al., 2022) | 7.42 | 0.22 | 24 | every 2 | 128 | 1 | 0 |
| LLaMA-MoE-v2-3.8B (Qu et al., 2024) | 8.03 | 3.80 | 32 | all | 8 | 2 | 0 |
| JetMoE-8B (Shen et al., 2024b) | 8.52 | 2.33 | 24 | all | 8 | 2 | 0 |
| OpenMoE-8B (Xue et al., 2024a) | 11.86 | 3.80 | 24 | every 6 | 32 | 2 | 1 |
| MiniCPM-MoE-8x2B (Hu et al., 2024) | 13.87 | 4.32 | 40 | all | 8 | 2 | 0 |
| Qwen1.5-MoE-A2.7B (Qwen Team, 2024) | 14.32 | 2.69 | 24 | all | 60 | 4 | 4 |
| DeepSeek-V2-Lite (DeepSeek-AI et al., 2024a) | 15.71 | 2.66 | 27 | after 1st | 64 | 6 | 2 |
| DeepSeekMoE (Dai et al., 2024) | 16.38 | 2.83 | 28 | after 1st | 64 | 6 | 2 |
| XVERSE-MoE-A4.2B (XVERSE Technology Inc., 2024) | 25.78 | 4.23 | 28 | all | 64 | 6 | 2 |
| Qwen3-30B-A3B (Yang et al., 2025) | 30.53 | 3.35 | 48 | all | 128 | 8 | 0 |
| Yuan2.0-M32 (Wu et al., 2024) | 39.94 | 3.70 | 24 | all | 32 | 2 | 0 |
| Phi-3.5-MoE (Abdin et al., 2024) | 41.87 | 6.64 | 32 | all | 16 | 2 | 0 |
| GRIN-MoE (Liu et al., 2024) | 41.87 | 6.64 | 32 | all | 16 | 2 | 0 |
| Mixtral-8x7B-v0.1 (Jiang et al., 2024) | 46.70 | 12.88 | 32 | all | 8 | 2 | 0 |
| Jamba-Mini-1.6 (Lenz et al., 2025) | 51.57 | 12.11 | 32 | every 2 | 16 | 2 | 0 |
| NLLB-MoE-54B (NLLB Team et al., 2022) | 54.50 | 3.75 | 48 | every 4 | 128 | 2 | 0 |
| Qwen2-57B-A14B (Yang et al., 2024a) | 57.41 | 14.25 | 28 | all | 64 | 8 | 8 |

- JetMoE-8B employs mixture-of-attention (Shen et al., 2024a), which we keep intact in our experiments.
- GRIN-MoE shares the same architecture with Phi-3.5-MoE, but is trained using different methods.
- Jamba-Mini-1.6 employs a hybrid SSM-Transformer structure, yet the MoE part is identical to vanilla transformer-based MoE models.

## D.2 TOY MODEL CONFIGURATIONS

To validate the potential factors that affects local routing consistency, we modify the configuration of OLMoE (Muennighoff et al., 2025) and create a series of toy MoE models which we pretrain from scratch. The baseline model, Baseline, has only 8 layers and a hidden dimension of 1,280, compared to the original 16 layers and a hidden dimension of 2,048. Other architectural hyperparameters, such as the number of experts (activate 8 out of 64 experts) and the hidden dimension ratio between attention and expert (2:1, so Baseline is 640 and the original OLMoE is 1,024), are left intact. We sample 20B tokens from OLMoE's pretraining data, and pretrained the model on it for 10,000 steps; other training configurations such as sequence length (4,096), global batch size (1,024) and learning rate (cosine decay from 4e-4 to 5e-5) all follow the original pretraining stage settings.

Starting from the configuration of Baseline, we tweak one single setting once to create the following TOY models (including Baseline); all models have around 1.43B parameters, although they may activate a different number of parameters:

- FewerExp: Use 32 experts instead of 64, with doubled expert hidden dimension (1,280) and halved activated experts (4).
- ActMore: Activate 16 experts instead of 8, under the same total number of experts;
- ActFewer: Activate 2 experts instead of 8, under the same total number of experts;
- 1ShrExp: Replace an expert with a shared expert, so the router selects 7 out of 63 experts;
- 2ShrExp: Replace 2 experts with shared experts, so the router selects 6 out of 62 experts;
- DenseFst: Replace the first layer with a dense MLP layer, whose hidden dimension is the sum of *all* experts' hidden dimensions (40,960);

- DenseHlf: Replace the 1st, 3rd, 5th and 7th layers with dense MLP layers same as of DenseFst;
- NoLB: Adjust the load balance auxiliary loss coefficient from 0.01 to 0 (no regularization);
- OverLB: Adjust the load balance auxiliary loss coefficient from 0.01 to 0.1 (over regularization).

### D.3 DATA PROCESSING AND INPUT GENERATION

We first extract samples from RedPajama and downstream application datasets in plain text format. For RedPajama, this is already done. For LMArena, each of the original instances consists of two human-LLM conversations and a preference vote. We keep the instances where one of the conversations is preferred and concatenate all rounds from the preferred conversation (each round with its role and content) into one document. For OpenMath, OpenCode, and OpenScience, we simply concatenate the input and output of each instance.

After collecting samples from each RedPajama category and the downstream application dataset, we concatenate the samples within the domain, cutting them into input sequences of 512 tokens (the context window size of SwitchTransformers). We sample 2,048 input sequences for each domain, resulting in 22,528 input samples in total.

For SwitchTransformers, since the model is trained for masked language modeling, we randomly select 64 tokens from each input sequence, masking them in the original sequence as the encoder input and constructing the corresponding label sequence as the decoder input. For NLLB-MoE, as the model is trained for machine translation, we use the same sequence (with the English language token prepended) as both the encoder input and the decoder input. All other models do not need further data preprocessing, as they are decoder-only and trained for next token prediction.

## E ADDITIONAL RESULTS

### E.1 BASE VS. POST-TRAINED

We selected three models—LLaMA-MoE-v1, OLMoE, and JetMoE—that have both base and post-trained versions released, and calculated SRP and $\hat{\rho}$ of each version. Table 6 lists the results, from which we can see that the differences of both SRP and $\hat{\rho}$ between models before and after post-training are not significant enough to change the degree of local routing consistency, regardless of what type of post-training (SFT, DPO, etc.) is applied. Another related fact is that Phi-MoE-3.5 and GRIN-MoE, which share the same model architecture but are trained differently, have similar local routing consistency. Both indicate that the training method may be less important than the model architecture concerning local routing consistency.

Table 6: SRP between models before and after post-training.

| Model | $m = 4$ | | $m = 16$ | | $m = 64$ | | $m = 256$ | |
|---|---|---|---|---|---|---|---|---|
| | SRP | $\hat{\rho}$ | SRP | $\hat{\rho}$ | SRP | $\hat{\rho}$ | SRP | $\hat{\rho}$ |
| LLaMA-MoE-v1 | 55.78 | 1.03 | 45.29 | 2.39 | 41.61 | 2.92 | 40.62 | 3.52 |
| +SFT | +0.01 | -0.00 | -0.01 | -0.00 | -0.01 | +0.00 | -0.00 | -0.00 |
| OLMoE | 64.69 | 1.00 | 50.91 | 1.06 | 45.53 | 1.21 | 42.64 | 1.19 |
| +SFT | +0.40 | +0.00 | +0.47 | +0.01 | +0.50 | +0.02 | +0.60 | -0.02 |
| +DPO | +0.37 | +0.00 | +0.43 | +0.01 | +0.47 | +0.02 | +0.57 | -0.02 |
| +Instruct | +0.45 | +0.00 | +0.56 | +0.01 | +0.62 | +0.02 | +0.74 | -0.02 |
| JetMoE | 60.22 | 1.09 | 47.45 | 2.26 | 42.78 | 2.69 | 41.09 | 3.15 |
| +SFT | -0.20 | -0.00 | -0.14 | +0.01 | -0.12 | +0.02 | -0.09 | +0.03 |
| +Chat | -0.20 | -0.00 | -0.15 | +0.01 | -0.13 | +0.02 | -0.10 | +0.03 |

### E.2 SRP PER SEGMENT POSITION

To determine whether the segment position $p$ can affect the segment routing best performance, we calculate SRP on each segment position by summarizing statistics of all segments that share the same

position. Figure 9 illustrates this position-wise SRP at each possible segment position. Most models have nearly constant SRP at every position except $p = 0$, where many models activate specialized experts to handle the beginning of the input sequence. This stability of local routing consistency across input positions allows us to use segments from all positions to calculate SRP, and apply conclusions based on SRP to any segment of the input (except the very first one).

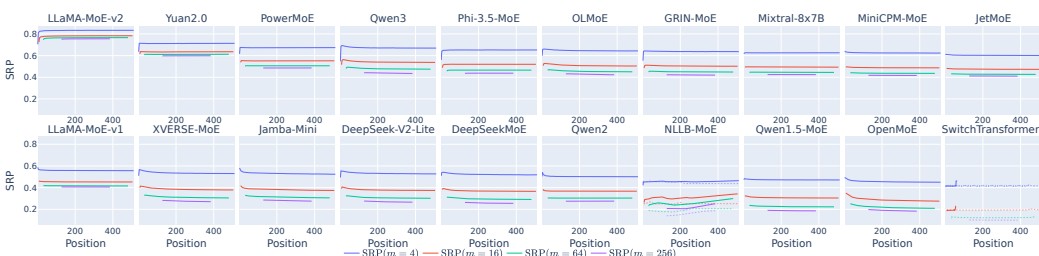

Figure 9: Position-wise SRP of each model on the full corpus. For encoder-decoder models, dotted lines show the encoder SRP and solid lines show the decoder ones.

### E.3 SRP ACROSS DOMAINS

To verify whether local routing consistency is transitive across different domains, we calculate the correlation of expert segment routing best performance between pair-wise domains and demonstrate it in Figure 10. We also compute the correlation of expert activation frequency between pair-wise domains, results illustrated in Figure 11. By comparing corresponding heapmaps, we can see that local routing consistency is nearly always positively correlated, even between distant domains on which the experts' activation frequencies are negatively correlated. This means that local routing consistency is transitive; domain-specialized experts with high local routing consistency in one domain tend to exhibit it in any other domain. We also found that some models (e.g., LLaMA-MoE-v2 and Qwen2) do not show a significant difference between domains, which is aligned with the results in Section 4.2.

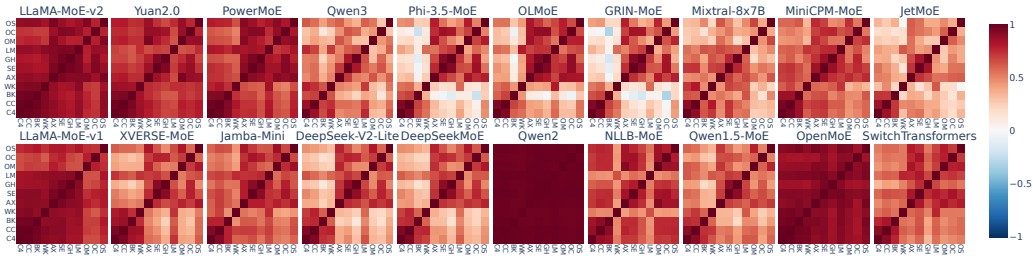

Figure 10: Correlation between domain-wise expert SRP of each model. C4: C4; CC: Common-Crawl; BK: Books; WK: Wikipedia; AX: ArXiv; SE: StackExchange; GH: GitHub; LM: LMArena; OM: OpenMath; OC: OpenCode; OS: OpenScience.

### E.4 SRP VS. SCH

To clarify the relation between SRP and SCH, Table 7 lists the correlation between them across all models. The two metrics are always highly positively correlated regardless of the values of $m$ and $\rho$. This ensures that SCH shares the same property of SRP under reasonable segment length and cache size. Furthermore, when $\rho$ is around 1.5, the two metrics are most closely related, nearly perfectly linear, aligned with Figure 4 where most models have $\hat{\rho} \in [1, 3]$ when $m \geq 16$, as well as our previous claim that $\rho = 2$ balances cache effectiveness and efficiency.

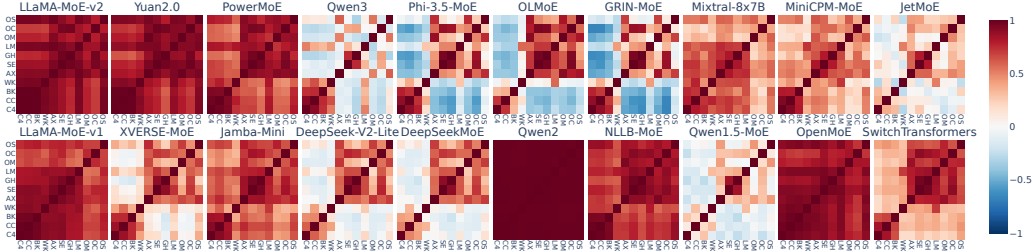

Figure 11: Correlation between the domain-wise expert activation frequency of each model. C4: C4; CC: CommonCrawl; BK: Books; WK: Wikipedia; AX: ArXiv; SE: StackExchange; GH: GitHub; LM: LMArena; OM: OpenMath; OC: OpenCode; OS: OpenScience.

Table 7: Correlation between SRP and SCH across all models. Bold font indicates the highest correlation across $\rho$ for each $m$.

| $m$ | $\rho = 0.5$ | $\rho = 1.0$ | $\rho = 1.5$ | $\rho = 2.0$ | $\rho = 2.5$ | $\rho = 3.0$ |
|---|---|---|---|---|---|---|
| 4 | 67.28 | 94.17 | **97.46** | 91.91 | 87.47 | 81.49 |
| 16 | 81.16 | 97.09 | **97.37** | 94.87 | 91.02 | 84.87 |
| 64 | 86.77 | 97.76 | **98.03** | 96.15 | 92.22 | 87.31 |
| 256 | 88.28 | 97.35 | **97.89** | 96.24 | 92.68 | 87.69 |

## E.5 STATISTICAL SIGNIFICANCE

Due to the very high correlation between SRP and SCH, we choose SCH to represent local routing consistency, and report the 95% confidence intervals when $\rho = 2$ in Table 8. The confidence intervals are obtained by bootstrapping 1,000 times with samples from the full corpus.

Table 8: 95% confidence interval of SCH ($\rho = 2$) of REAL models. The decoder of SwitchTransformer does not have valid data when $m = 256$.

| Model | $m = 4$ | $m = 16$ | $m = 64$ | $m = 256$ |
|---|---|---|---|---|
| LLaMA-MoE-v2 | $(96.88, 97.17)$ | $(97.82, 98.03)$ | $(97.28, 97.58)$ | $(96.64, 97.01)$ |
| Yuan2.0 | $(77.45, 78.17)$ | $(81.69, 82.33)$ | $(78.69, 79.44)$ | $(76.98, 77.83)$ |
| PowerMoE | $(79.77, 80.30)$ | $(80.62, 81.24)$ | $(74.86, 75.71)$ | $(72.47, 73.44)$ |
| Qwen3 | $(72.86, 73.78)$ | $(76.49, 77.46)$ | $(68.14, 69.46)$ | $(62.91, 64.55)$ |
| Phi-3.5-MoE | $(72.81, 74.03)$ | $(74.57, 75.91)$ | $(67.37, 69.13)$ | $(63.65, 65.71)$ |
| OLMoE | $(71.27, 72.46)$ | $(73.39, 74.71)$ | $(65.74, 67.42)$ | $(61.94, 63.87)$ |
| GRIN-MoE | $(71.02, 72.24)$ | $(72.75, 74.09)$ | $(65.35, 67.10)$ | $(61.53, 63.57)$ |
| Mixtral-8x7B | $(76.67, 77.03)$ | $(73.92, 74.42)$ | $(66.59, 67.20)$ | $(63.16, 63.86)$ |
| MiniCPM-MoE | $(76.14, 76.43)$ | $(73.29, 73.67)$ | $(65.57, 65.96)$ | $(61.99, 62.44)$ |
| JetMoE | $(74.37, 74.66)$ | $(70.60, 71.00)$ | $(63.60, 64.04)$ | $(60.30, 60.80)$ |
| LLaMA-MoE-v1 | $(70.50, 70.80)$ | $(66.07, 66.44)$ | $(60.64, 61.02)$ | $(58.16, 58.57)$ |
| XVERSE-MoE | $(55.32, 56.04)$ | $(58.38, 59.14)$ | $(47.53, 48.40)$ | $(42.72, 43.64)$ |
| Jamba-Mini-1.6 | $(57.33, 58.00)$ | $(57.51, 58.24)$ | $(47.49, 48.37)$ | $(42.82, 43.84)$ |
| DeepSeek-V2-Lite | $(55.16, 56.00)$ | $(57.65, 58.49)$ | $(47.04, 47.96)$ | $(41.98, 42.99)$ |
| DeepSeekMoE | $(53.77, 54.65)$ | $(56.24, 57.13)$ | $(45.41, 46.52)$ | $(40.25, 41.28)$ |
| Qwen2 | $(54.73, 55.09)$ | $(54.75, 55.08)$ | $(46.52, 46.74)$ | $(42.91, 43.14)$ |
| NLLB-MoE (en) | $(28.36, 29.55)$ | $(35.92, 36.96)$ | $(28.94, 30.06)$ | $(24.62, 25.79)$ |
| (de) | $(35.39, 36.87)$ | $(42.26, 43.54)$ | $(37.31, 38.61)$ | $(32.51, 33.83)$ |
| Qwen1.5-MoE | $(40.67, 41.47)$ | $(45.79, 46.50)$ | $(34.73, 35.50)$ | $(29.45, 30.32)$ |
| OpenMoE | $(35.21, 36.55)$ | $(39.42, 40.68)$ | $(32.35, 33.80)$ | $(29.02, 30.56)$ |
| SwitchTF (en) | $(12.39, 13.58)$ | $(20.74, 21.86)$ | $(16.92, 18.06)$ | $(14.50, 15.66)$ |
| (de) | $(16.43, 17.81)$ | $(23.89, 25.16)$ | $(21.39, 22.61)$ | $(21.39, 22.61)$ |

## E.6 LAYER LEVEL RESULTS

Figure 12 illustrates each model's layer-wise SRP. Most models have peak SRPs among middle layers, while some (e.g., Yuan2.0 and MiniCPM) have another peak at the last layer. We conjecture that middle layers are less tied to input/output tokens and thus more sensitive to the general topic, and the final layers process highly abstract information that is also more related to the overall topic. Both encourage routers to select similar experts within a local segment that share the same topic across tokens. PowerMoE and Qwen2 have another peak on layer 2 due to expert imbalance. Appendix E.7 gives a clear view on this.

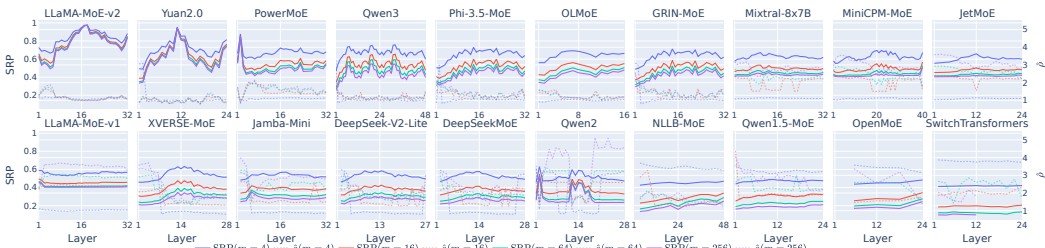

Figure 12: Layer-wise SRP on the full corpus of each REAL model. Solid lines show SRP while dotted lines show corresponding $\hat{\rho}$.

We also calculated layer-wise SCH, results demonstrated in Figure 13. The patterns are the same as SRP, indicating a high correlation between the two metrics.

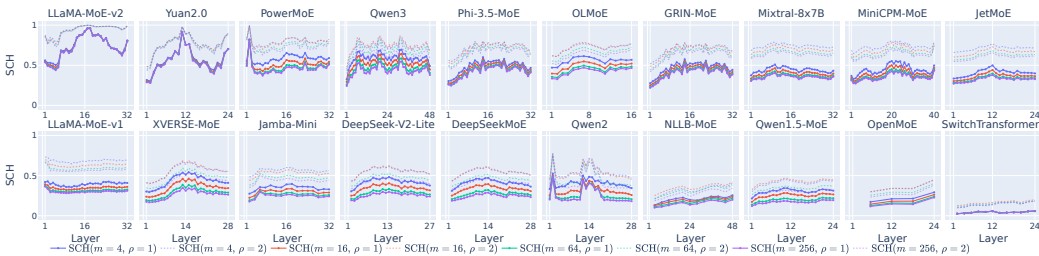

Figure 13: Layer-wise SCH on the full corpus of each REAL model. Solid lines show SCH when $\rho = 1$ while dotted lines show SCH when $\rho = 2$.

## E.7 EXPERT LEVEL RESULTS

We demonstrate expert-wise segment routing best performance against activation frequency in Figure 14. LLaMA-MoE-v2, Yuan2.0, and PowerMoE have experts with very high activation frequency. These experts naturally have very high local routing consistency and contribute to these models' high model-level local routing consistency. The imbalanced experts of PowerMoE mainly belong to layer 2, which also explains the observation in Section E.6.

Furthermore, Figures 15, 16, 17 and 18 compares SRP with domain and vocabulary specialzations. The plots are aligned with the conclusion of Section 4.2 that when the model exhibits domain specialization, domain-specialized experts contribute more to overall local routing consistency than vocabulary-specialized experts.

## F FURTHER DISCUSSIONS ON THROUGHPUT

The purpose of SRP and SCH is to provide a general metric that is agnostic to concrete implementations, where SRP solely depends on the model and SCH also considers a hard limit on cached experts. On the other hand, the performance of a true expert offloading system, usually measured through throughput, is not only decided by the deployed model, but also the implementation of the

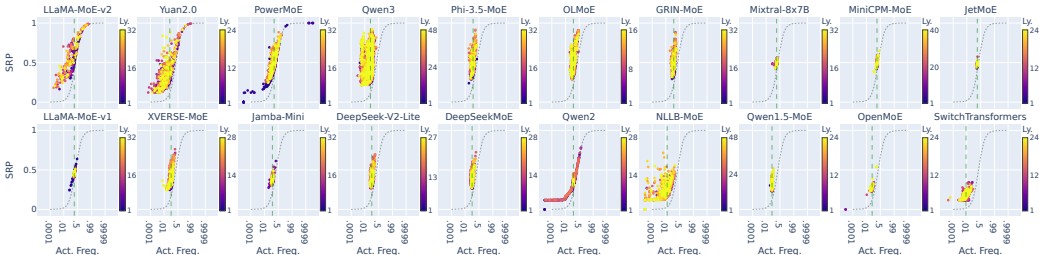

Figure 14: Per-expert activate frequency vs. SRP. The x-axis is stretched to show experts with very low or high activation frequency. Gray dashed lines indicate the theoretical lower bound of SRP at different activation frequencies. Green dashed lines show the expected activation frequency of experts from each model.

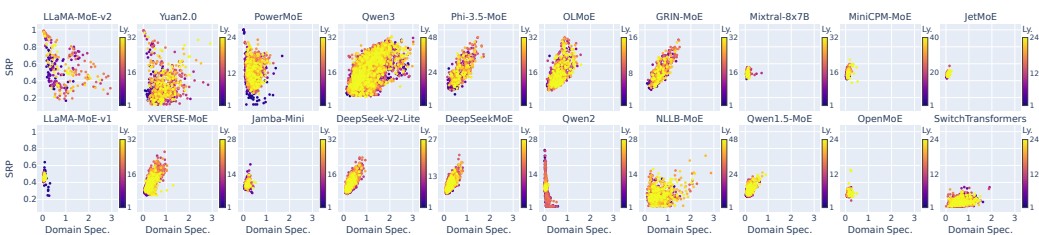

Figure 15: Per-expert domain specialization vs. SRP.

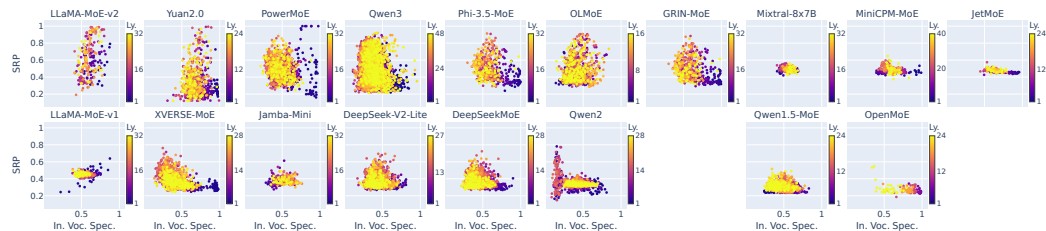

Figure 16: Per-expert input vocabulary specialization vs. SRP. Encoder-decoder models are not involved due to different input formats from other decoder-only models.

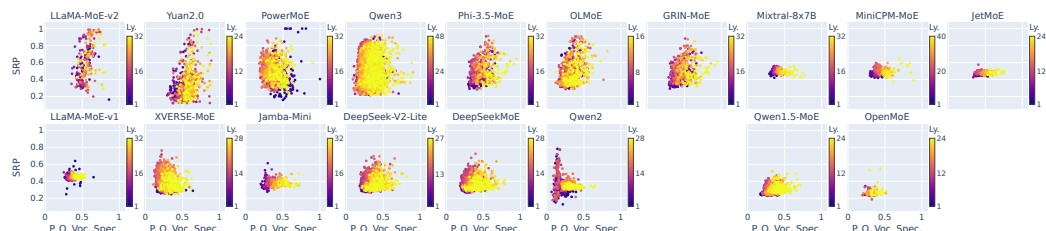

Figure 17: Per-expert predicted output vocabulary specialization vs. SRP. Encoder-decoder models are not involved due to different input formats from other decoder-only models.

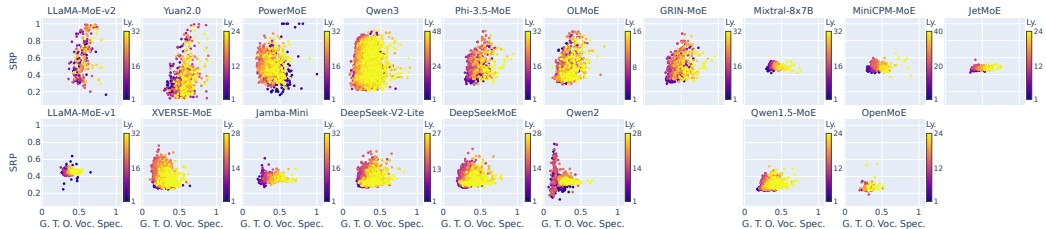

Figure 18: Per-expert ground-truth output vocabulary specialization vs. SRP. Encoder-decoder models are not involved due to different input formats from other decoder-only models.

system, including but not limited to cache management, overlap exploitation, hardware coordinate, etc. Nevertheless, local routing consistency still plays an important role on the model side. Below is an informal, theoretical analysis about how local routing consistency (we use SCH as an example) may affect the actual throughput.

For simplicity, we assume that there is only one GPU with limited GPU memory (insufficient for the whole model but enough for activated parameters and calculation), and a group of CPU with sufficient flash memory. (This is a common configuration on edge devices.) Consider an expert offloading system that offloads whole experts only. During the decoding stage (the more time-consuming stage), compared to full GPU inference, it may introduce the following overhead:

1. During the calculation of the last layer, the system (if capable) may predict what experts the upcoming layer (or the next forward run) will use, and prefetch these experts to GPU. The overhead of prefetching one expert can be relatively small because the prefetch process can overlap with the current calculation.

2. After the router decides what experts to use, if a demanded expert is not on the GPU, the system will need to either (1) load the expert to GPU on-the-fly, adding a communication overhead, or (2) run the expert on CPU directly, adding a calculation overhead. Both overheads are more significant than the prefetch overhead because no overlap can be utilized.

Based on the above analysis, during a forward run in the decoding phase, an ideal expert offloading system will always prefetch the correct group of experts for the upcoming layer or the next forward run, so the only overhead occurs during prefetching. This overhead also consists of two parts: (1) predicting the prefetched experts, whose overhead can be treated as constant as the system is ideal; (2) loading the selected experts, whose overhead is proportional to the number of cache misses between forward runs. As long as the GPU memory can hold more experts than the activated ones, the system will have to decide what extra experts to keep on GPU. When the expert activation sequence is known, the optimal eviction list is given by the Beladi algorithm; however, this algorithm relies on the precise time each expert will be activated in the future, which is very difficult to predict in practice. To this end, SCH with a specific segment length can be used as an approximation that considers the frequency of close-future expert activations, which is easier to predict. Therefore, SCH (more precisely $1 - \text{SCH}$) can be seen as an upper bound of the minimum number of cache misses, which is approximately proportional to the minimum overhead any expert caching system under the same single-expert prefetching overhead.

To further verify the relation between local routing consistency and the actual throughput or overhead of typical expert offloading systems, we implemented a naive version that utilizes LRU cache and always load missed experts on-demand. We use it to deploy all TOY models and measure their throughput on the full corpus under different cache sizes. The benchmark results show that the relative overhead w.r.t. full GPU inference has different correlations to SCH at the two inference stages: positive during prefilling ($r \approx 0.2$), and negative during decoding ($r \approx -0.3$). The relation holds under various cache size ratio $\rho$. Note that the directions of the correlations align with the relation between local routing consistency and local load balance: During prefilling, multiple consecutive tokens are processed in one run, where tokens belong to the same expert will be dispatched to that expert together, so the bottleneck is the largest number of tokens an expert needs to process, which is related to local load balance. During decoding, however, only one token is processed per run, making the activated expert distribution between consecutive runs more important, which is the concern of local routing consistency. Since decoding is almost always more time-consuming on single queries,

we conclude that local routing consistency is more important and local load balance may be sacrificed to some extent. Nevertheless, the correlation coefficient is not far from 0, indicating that there are also other significant factors that affects the system throughput, so the model (as well as local routing consistency) should not be the sole decider.

