# OpenReview forum: "Not All Models Suit Expert Offloading: On Local Routing Consistency of Mixture-of-Expert Models"
_ICLR.cc/2026/Conference — ICLR 2026 Poster_

### Official Review · Reviewer_3aLj · 2025-10-31

**Soundness:** 3
**Presentation:** 1
**Contribution:** 3
**Rating:** 6
**Confidence:** 3

**Summary:**

MoE architectures have recently enabled scaling of LLMs to even larger sizes without introducing the same amount of computation. To reduce the memory footprint, expert offloading has become a major strategy for inference under memory-constrained environments. However, limited studies have examined the potential cache hit rate of different MoE LLMs in recent progress. The authors performed experiments across a wide range of MoE LLMs and datasets and identified that routing consistency can vary across models. Based on this observation, the authors proposed several metrics to estimate the capability of a particular MoE model to perform expert offloading.

**Strengths:**

- The idea is intuitive and based on an interesting observation that the activation consistency could change due to MoE architectures. Early studies in this field were unable to cover the later architectural changes in MoEs, so the proposed study addresses an area that has not been well studied.

- The selected models and datasets are comprehensive, covering both classic and recent variants of MoE LLMs (within hardware constraints). Based on these empirical results, the authors have provided some great observations.

**Weaknesses:**

- The abstract is poorly written. It does not clearly mention the different MoE architectures (shared experts, small/large experts) as the potential reasons for the local consistency discrepancy, which I believe is the most important contribution of this paper.

- While the authors have shown that the different MoE models could have vastly different expert activation consistency, the authors did not perform an end-to-end empirical evaluation on the expert offloading performance with the proposed strategy. As the MoE model architectures themselves could be vastly different, whether a certain model suit expert offloading would not only depends on the cache hit rate.

- I personally believe that it would be better to describe the concepts of SRP and SCH using visualizations. While the listed formulas are helpful, they do not effectively convey the core idea. For SCH in particular, since it describes the cache hit rate, the authors should consider including a figure to illustrate the concept, as is common in other papers that discuss caching.

- Figure 2-6 are way too small. It's really difficult for me to identify the relevant colors even with a large screen. These figures give me the sense that they were copy pasted from some research reporting slides, instead of being curated for this manuscript.
The author should consider ways to improve the rendering of these figures, especially for figure 2 and 3, given that these figure consist of the most important observation of this paper.

- Conclusion is badly written, as it does not make any suggestion on the MoE architecture as promised by the abstract.

- Minor
  - L150 -- single expert is a weird paragraph title here. "Single expert case" might be better as it's mostly related to SRP, instead of any particular expert. Similar comment could be made to line 180 and 256 ("Dataset").

**Questions:**

I would like to see some of the discussion regarding the actual performance of offloading that takes the architectural differences into the consideration.

It would also be interesting to see why particular training strategy or model architecture design leads to the split between the domain specialization / vocabulary specialization in the MoE models.

---

> ### Author Response · Authors · 2025-12-03
>
> ## A1: Paper organization
>
> > **W1 & W5:** The abstract does not clearly mention the different MoE architectures (shared experts, small/large experts) as the potential reasons for the local consistency discrepancy, which I believe is the most important contribution of this paper. Conclusion also does not make any suggestion on the MoE architecture as promised by the abstract.
>
> Thank you for your kind suggestion about the paper organization. In the revised manuscript, we will adjust the abstract to emphasize the insights about model architecture; we will also rearrange Section 2 to make space for more details in the conclusion.
>
> ## A2: Throughput experiments
>
> > **W2 & Q1:** While the authors have shown that the different MoE models could have vastly different expert activation consistency, the authors did not perform an end-to-end empirical evaluation on the expert offloading performance with the proposed strategy. As the MoE model architectures themselves could be vastly different, whether a certain model suit expert offloading would not only depends on the cache hit rate. I would like to see some of the discussion regarding the actual performance of offloading that takes the architectural differences into the consideration.
>
> We agree that the actual throughput of a certain model with expert offloading depends not only on its routing decisions but also on other factors such as model size and sparsity; however, this also makes direct comparison between the throughput of existing MoE models incomprehensible. Furthermore, the concrete implementation of expert offloading also affects the throughput.
>
> Nevertheless, we conduct experiments on standard toy models based on MoE that have comparable sizes (all 1.43B) and controlled MoE designs, benchmarking their throughput with a naive expert offloading system with LRU cache. We use the extra overhead to measure how "suitable" a model is for expert offloading system: if the overhead is small then expert offloading is efficient on this model. **Table 1** lists the correlation between the relative offloading overhead and average LRU hit rate under different cache size ratio $\rho$. We can see that in the decoding phase, offloading overhead is negatively correlated to LRU hit rate; the prefilling phase is the opposite case, which matches the conclusion about local routing consistency and local load balance in Section 3.3. We also observe that some other factors can affect the relative offloading overhead: for example, if the model activates very few parameters, then even if cache misses happen all the time, each time the extra load is small and thus the overhead is not large. Nevertheless, we argue that cache hit rate (including SCH) can somehow reflect expert offloading performance.
>
> | $\rho$ | Prefill |  Decode |
> | :----: | :-----: | :-----: |
> |    1.0 |  0.1618 | -0.1777 |
> |    2.0 |  0.2987 | -0.3151 |
> |    3.0 |  0.1898 | -0.3379 |
>
> > **Table R1:** Correlation between offloading overhead and average LRU hit rate measured on the full corpus, under various cache size ratio $\rho$.

---

> > ### Author Response · Authors · 2025-12-03
> >
> > ## A3: Visualization and presentation
> >
> > > **W3:** I personally believe that it would be better to describe the concepts of SRP and SCH using visualizations. While the listed formulas are helpful, they do not effectively convey the core idea. For SCH in particular, since it describes the cache hit rate, the authors should consider including a figure to illustrate the concept, as is common in other papers that discuss caching.
> >
> > Thank you for your kind suggestion about a visualized introduction to the metrics. We will rearrange Section 2 to provide a clearer and more intuitive illustration and move some equations to the Appendix in the revised manuscript.
> >
> > > **W4:** Figure 2-6 are way too small. It's really difficult for me to identify the relevant colors even with a large screen. These figures give me the sense that they were copy pasted from some research reporting slides, instead of being curated for this manuscript. The author should consider ways to improve the rendering of these figures, especially for figure 2 and 3, given that these figure consist of the most important observation of this paper.
> >
> > Thank you for your suggestion on the figures. We will enlarge the details in these figures and move some sub-figures to the Appendix in our revised manuscript.
> >
> > > **W6:** L150 -- single expert is a weird paragraph title here. "Single expert case" might be better as it's mostly related to SRP, instead of any particular expert. Similar comment could be made to line 180 and 256 ("Dataset").
> >
> > Thank you for pointing out potential improvements in the wording; we will adjust them in our revised manuscript. Meanwhile, we would also like to point out that SRP *can* apply to individual experts, which we utilized when comparing SRP and expert specialization.
> >
> > ## A4: Expert specialization and model architecture
> >
> > > **Q2:** It would also be interesting to see why particular training strategy or model architecture design leads to the split between the domain specialization / vocabulary specialization in the MoE models.
> >
> > We understand that expert specialization is important in MoE models and we agree that knowing the driving factors of such phenomenon is beneficial for future development of MoE models; however, this topic is slightly out of scope for our work, and we believe that a future work that fully investigate expert specialization will be more appropriate. Despite so, we measure the average expert specialization on the standard toy models introduced in A2, results shown in **Table R2**. The most significant factor is load balance regularization: without it, expert specialization (especially domain specialization) grows significantly, similar to LLaMA-MoE-2 and Yuan. Meanwhile, activating more experts decreases expert specialization, which is intuitive because each expert is activated more frequently, potentially under more topics. Other architectural modifications do not bring significant expert specialization fluctuations.
> >
> > | Model    | Domain | Input  | Pred.  | G.T.   |
> > | :------: | :----: | :----: | :----: | :----: |
> > | Baseline | 0.4744 | 0.6065 | 0.3796 | 0.3231 |
> > | FewerExp | 0.4386 | 0.6909 | 0.4397 | 0.3794 |
> > | ActMore  | 0.3859 | 0.5686 | 0.3748 | 0.3185 |
> > | ActFewer | 0.5443 | 0.7494 | 0.3969 | 0.3338 |
> > | 1ShrExp  | 0.4693 | 0.6226 | 0.3799 | 0.3222 |
> > | 2ShrExp  | 0.4670 | 0.6460 | 0.3778 | 0.3194 |
> > | DenseFst | 0.5189 | 0.5119 | 0.3581 | 0.3017 |
> > | DenseHlf | 0.4702 | 0.6376 | 0.3916 | 0.3267 |
> > | NoLB     | 0.6205 | 0.6225 | 0.4105 | 0.3578 |
> > | OverLB   | 0.2393 | 0.6461 | 0.3645 | 0.3097 |
> >
> > > **Table R2:** Expert specialization of OLMoE-based toy models; "Input", "Pred." and "G.T." columns show the three types of vocabulary specialization (input, model predicted output and ground-truth). Detailed settings are covered in the revised manuscript.

---

### Official Review · Reviewer_9czm · 2025-10-31

**Soundness:** 2
**Presentation:** 3
**Contribution:** 2
**Rating:** 4
**Confidence:** 3

**Summary:**

This paper investigates a key property of Mixture-of-Experts (MoE) models, termed "local routing consistency", which determines a model's suitability for expert offloading systems. To quantify this property, the paper proposes two novel metrics: Segment Routing Best Performance (SRP) and Segment Cache Best Hit Rate (SCH). Through an extensive empirical study on 20 MoE large language models with diverse architectures, the authors obtain several findings, for example, models applying MoE on every layer and without shared experts exhibit the highest routing consistency.

**Strengths:**

This paper offers a novel analytical perspective on the important and practical problem of efficient MoE model deployment. The concept of "local routing consistency" is insightful. It provides a clear and quantifiable framework for evaluating and comparing the deployment potential of different MoE models in resource-constrained environments.

The experimental evaluation is thorough and comprehensive, standing out as a primary strength of this work. The authors analyze up to 20 representative MoE models, covering a wide range of parameter scales and architectural variants, which lends strong credibility and generalizability to their conclusions.

**Weaknesses:**

Although the analysis is insightful, its conclusions are primarily based on correlation rather than causation. For example, the study observes that architectures with "MoE on every layer" and "no shared experts" correlate with high routing consistency and conjectures that dense modules might "interfere with or weaken routing signals". Ablation studies, such as modifying these architectural features on the same backbone (even a small one is ok) and observing the change in consistency, would greatly strengthen the reliability of these claims and uncover the underlying mechanisms.

The work did not explore the potential trade-off between this routing consistency and the model's core performance (e.g., accuracy). It is plausible that forcing or guiding a model to produce highly consistent routing could limit its expressive power, making it harder to capture fine-grained, token-level semantic shifts and thus hurting its prediction accuracy. The paper lacks a discussion on this "efficiency-accuracy" trade-off, which makes it difficult for readers to assess whether the design choices made to pursue high SRP/SCH might come at an unacceptable performance cost.

The paper's definition of "local routing consistency" primarily focuses on the processing of contiguous text segments, which corresponds mainly to the autoregressive decoding phase. However, in practical applications, the prefilling phase for long prompts is also a significant performance bottleneck. The routing patterns during prefilling may differ from those during decoding, but the current analytical framework does not explicitly distinguish between or investigate consistency in these two modes.

The paper's core claim is to guide memory-efficient model design and deployment, yet this claim is not well demonstrated. The study focuses primarily on analyzing proxy metrics like cache hit rates but does not provide implementation of a real model or a real expert offloading system.

**Questions:**

1. Could you design an experiment to more directly verify that dense components indeed "interfere" with sparse routing, thereby decreasing consistency?

2. Is there an inherent trade-off between maximizing local routing consistency (for inference efficiency) and maintaining the model's predictive performance (e.g., accuracy)? Could designs that pursue high SRP potentially limit the model's expressive capacity?

3. Does your analysis and conclusions apply equally to the prefill stage for long context inputs?

4. How well do the findings in this paper guide model or system design? Specifically, is there any action item (with validated performance) for the MoE model/system developers to optimize their design?

---

> ### Author Response · Authors · 2025-12-03
>
> ## A1: Model architecture
>
> > **W1 & Q1**: Although the analysis is insightful, its conclusions are primarily based on correlation rather than causation. For example, the study observes that architectures with "MoE on every layer" and "no shared experts" correlate with high routing consistency and conjectures that dense modules might "interfere with or weaken routing signals". Could you design an experiment to more directly verify that dense components indeed "interfere" with sparse routing, thereby decreasing consistency?
>
> We agree that the observational results do not fully reveal the relation between model architecture and local routing consistency. In order to further examine our conjectures, we conduct experiments on standard toy models based on OLMoE that have comparable sizes (all 1.43B) and controlled MoE designs. **Table R1** illustrates parts of the results when segment length $m=16$, where we find that adding dense *layers* actually increases SRP a little, but with the cost of a higher perplexity. However, adding another dense component, *shared experts*, significantly decreases SRP and does not lower perplexity. Based on the results, we propose two hypothesis *specifically* for the effects of shared experts: bypass effect, where shared experts process more information than the original group of experts; shrinked expert combination space, where routers have less expert combination choices for different inputs due to decreased number of total and activated experts. We will include the full results and an updated Section 3.3 in the revised manuscript.
>
> | Model    | SRP   | log PPL |
> | :------: | :---: | :-----: |
> | Baseline | 43.56 |    2.40 |
> | 1ShrExp  | 41.38 |    2.40 |
> | 2ShrExp  | 38.78 |    2.41 |
> | DenseFst | 44.87 |    2.43 |
> | DenseHlf | 43.67 |    2.46 |
>
> > **Table R1:** SRP ($m=16$) and log perplexity on the full corpus of several standard toy models. 1ShrExp and 2ShrExp replace 1 and 2 experts to shared experts respectively; DenseFst replaces the first layer with a dense MLP; DenseHlf does so every two layers.
>
> ## A2: Language modeling performance
>
> > **W2 & Q2**: The work did not explore the potential trade-off between this routing consistency and the model's core performance (e.g., accuracy). It is plausible that forcing or guiding a model to produce highly consistent routing could limit its expressive power, making it harder to capture fine-grained, token-level semantic shifts and thus hurting its prediction accuracy. Is there an inherent trade-off between maximizing local routing consistency (for inference efficiency) and maintaining the model's predictive performance (e.g., accuracy)? Could designs that pursue high SRP potentially limit the model's expressive capacity?
>
> In Appendix E.2, we already illustrate the relation between SRP and model perplexity (which reflects some level of LM performance), where we do not find significant correlation. We also conduct the same experiments on the standard toy models introduced above: From **Table R1** we can see that a high SRP does not equal to a high perplexity; although some radical modification (e.g., dropping load balance regulation) for high SRP does harm LM perfomrance, in general we do not find a strong negative connection between SRP and LM performance on the toy models.
>
> Besides empirical results, in Section 4 we conclude that *domain specialized* experts have main contribution to local routing consistency; models can have a (large) group of domain specialized experts for consisitent routing, along with a (small) group of vocabulary specialized experts to handle "fine-grained, token-level semantic shifts" (an example being Qwen3).
>
> Finally, we would like to argue that local routing consistency is one of the trade-off aspects alongside with LM performance, load balance, etc.; it may act as a regularization but shall not dominate the core optimization target of LLMs.

---

> > ### Author Response · Authors · 2025-12-03
> >
> > ## A3: Prefilling stage analysis
> >
> > > **W3 & Q3:** The paper's definition of "local routing consistency" primarily focuses on the processing of contiguous text segments, which corresponds mainly to the autoregressive decoding phase. However, in practical applications, the prefilling phase for long prompts is also a significant performance bottleneck. The routing patterns during prefilling may differ from those during decoding, but does your analysis and conclusions apply equally to the prefill stage for long context inputs?
> >
> > During prefilling, the model calculates routing decisions for multiple tokens; for long prompts this usually results in involving **most or all experts** in a single forward run. To maximize batch calculation, inference systems will dispatch all tokens that activate a certain expert to that expert and calculate in one shot. Therefore, we believe that **(local) load balance** is more important than **local routing consistency** in the prefilling phase for long inputs as the bottleneck mainly depends on the largest number of tokens one expert has to handle.
> >
> > ## A4: Concrete implementation
> >
> > > **W4 & Q4:** The paper's core claim is to guide memory-efficient model design and deployment, yet this claim is not well demonstrated. The study focuses primarily on analyzing proxy metrics like cache hit rates but does not provide implementation of a real model or a real expert offloading system. How well do the findings in this paper guide model or system design? Specifically, is there any action item (with validated performance) for the MoE model/system developers to optimize their design?
> >
> > The main focus of our work is to inspect how suitable an MoE model can be for expert offloading in the general case; we do not intend to propose a specific offloading system or limit our conclusions to such one. As for model architecture design, the toy experiments in A1 can be seen as a concrete example, and we look forward to future studies that scale up these models to further examine our insights.

---

### Official Review · Reviewer_jCkR · 2025-11-05

**Soundness:** 2
**Presentation:** 2
**Contribution:** 3
**Rating:** 4
**Confidence:** 4

**Summary:**

Mixture-of-expert (MoE) models typically have high parameter count and not all parameters are always present on the GPUs. Efficiency and throughput of the model will be higher if active experts are present in the GPUs and there is minimal computation on CPU / communication between CPU and GPU. This work proposes two metrics to analyze how correlated expert activations are within a segment of tokens (termed local routing consistency). The authors empirically analyze various MoE models and suggest architecture design choices that might impact the local consistency. They also propose a way to empirically select the ideal cache size for a given model.

**Strengths:**

The authors tackle a specific and important problem in LLM deployment. The work proposes two metrics (SRP and SCH) to analyze the local routing consistency. The empirical analysis is quite thorough with 20 different MoE models and several datasets from both general training corpora and downstream tasks. The paper considers several reasons that could be responsible for the consistency - model architecture, expert specialization to specific domains or vocabulary subsets and load balancing. The findings are interesting - shared experts have lower local consistency (SRP), load-balanced models can have high SRP and cache size of $2\times$ number of experts results in a high SCH for most models.

**Weaknesses:**

1. There is no explicit connection of proposed metrics to throughput. For instance, how would SRP/SCH affect throughput given cache size, communication time and LLM forward propagation time? There are no measurements of throughput in any of the experiments either.
2. What advantage does SCH have compared to common cache algorithm hit rate? Can we not determine required cache size by analyzing LRU hit rate vs $\rho$?
3. In analysis in Section 3.3, the authors suggest that applying MoE on every layer and not sharing experts results in higher routing consistency. While there might be a correlation between the two, it does not imply causation. Particularly in the case of ‘MoE on every layer’, there are just 4 out of 20 models that apply at every ‘k’ layer instead and of those four, two of them have extremely high ratio of active to total experts (1:64 and 1:128). It is difficult to come to such conclusions without decoupling the factors and with such limited data. The conjecture on why these might cause inconsistency (L:334-336) is also unclear.
4. Writing needs significant improvement: \
i. The proposed metrics in Section 2 require a lot of time to understand. Providing intuitive / textual explanations for the equations, removing repeated mentioning of variables (for e.g., saying expert $e \in E$ at all locations instead of just saying expert) and moving unnecessary equations to appendix and replacing them with text in the main section would vastly improve the reader’s experience. \
ii. The term $\rho$ is overloaded - it is used in both SRP for segment routing size ratio and in SCH for segment cache size ratio - causing a lot of confusion. \
iii. Some of the figures (e.g. Figures 2 and 4) are hard to parse.
5. Since the proposed metric requires access to expert activation, it is not possible to measure them for black box models.

**Questions:**

1. Questions are primarily based on the weaknesses above. Provide answers to weaknesses (1) and (2). If possible provide a function connecting the proposed metrics to throughput / inference time and empirical results for the same.
2. Clearly explain why the dense modules might decrease local consistency (SRP).
3. Is it useful to have the marker size representing model size in Figure 2? Similarly, can expert specialization vs SRP be used in the plot in Figure 4 instead of correlation? Why do LL2 and Y2 have a very low correlation while they have high SRP and relatively high expert domain specialization? Why do STe have high correlation but low domain specialization?

---

> ### Author Response · Authors · 2025-12-03
>
> ## A1: Throughput experiments
>
> > **W1 & Q1:** There is no explicit connection of proposed metrics to throughput. For instance, how would SRP/SCH affect throughput given cache size, communication time and LLM forward propagation time? There are no measurements of throughput in any of the experiments either. If possible provide a function connecting the proposed metrics to throughput/inference time and empirical results for the same.
>
> We agree that throughput is a more straightforwoard indicator to compare between models that are actually deployed for inference, however it must rely on some concrete expert offloading implementations. The purpose of SRP and SCH is to provide a general metric that is agnostic to such implementations, where SRP solely depends on the model and SCH also considers a hard limit on cached experts. Here we provide a theoretical analysis to explain how SCH may affect the actual throughput, as this metric is closer to real expert offloading settings.
>
> ---
>
> For simplicity, we assume that there is only one GPU with limited GPU memory (insufficient for the whole model but enough for activated parameters and calculation), and a group of CPU with sufficient flash memory. (This is a common configuration on edge devices.) Consider an expert offloading system that offloads whole experts only. During the decoding stage (the more time-consuming stage), compared to full GPU inference, it may introduce the following overhead:
>
> 1. During the calculation of the last layer, the system (if capable) may predict what experts the upcoming layer (or the next forward run) will use, and prefetch these experts to GPU. The overhead of prefetching one expert can be relatively small because the prefetch process can overlap with the current calculation.
>
> 2. After the router decides what experts to use, if a demanded expert is not on the GPU, the system will need to either (1) load the expert to GPU on-the-fly, adding a communication overhead, or (2) run the expert on CPU directly, adding a calculation overhead. Both overheads are more significant than the prefetch overhead because no overlap can be utilized.
>
> Based on the above analysis, during a forward run in the decoding phase, an ideal expert offloading system will always prefetch the correct group of experts for the upcoming layer or the next forward run, so the **only overhead occurs during prefetching**. This overhead also consists of two parts: (1) predicting the prefetched experts, whose overhead can be treated as constant as the system is ideal; (2) loading the selected experts, whose overhead is **proportional to the number of cache misses between forward runs**. As long as the GPU memory can hold more experts than the activated ones, the system will have to decide what extra experts to keep on GPU. When the expert activation sequence is known, the optimal eviction list is given by the Beladi algorithm; however, this algorithm relies on the precise time each expert will be activated in the future, which is very difficult to predict in practice. To this end, SCH with a specific segment length can be used as an approximation that considers the frequency of close-future expert activations, which is easier to predict. Therefore, SCH (more precisely $1-\mathrm{SCH}$) can be seen as an **upper bound of the minimum number of cache misses, which is approximately proportional to the minimum overhead any expert caching system under the same single-expert prefetching overhead.** (Note: this also applies to systems that only make on-demand loads by replacing prefetching overhead with on-demand loading overhead.)
>
> ---
>
> The current definition of SCH does not take cache misses between segments into account. To fix this, we apply a slight modification: now the oracle cache loads demanded experts at every token, evicting unused ones based on the frequency of future expert activations within a period. The results and conclusions are similar, and will be updated in the revised manuscript.

---

> > ### Author Response · Authors · 2025-12-03
> >
> > Our analysis is general and does not involve any concrete offloading designs. Nevertheless, we measure the throughput of standard toy models introduced in A3 (including the ones in **Table R2** and other variants such as fewer experts and no load balance regularization) with a naive expert offloading system with LRU cache, and compare the introduced overhead (relative to full GPU inference time) with the average LRU hit rate. **Table R1** lists their correlation coefficient under different stages, where we can see that in the decoding phase, offloading overhead is negatively correlated to LRU hit rate; the prefilling phase is the opposite case, which matches the conclusion about local routing consistency and local load balance in Section 3.3. Based on these results, we argue that (average) cache hit rate can somehow reflect expert offloading performance. We will include more experiment details in the revised manuscript.
> >
> > | $\rho$ | Prefill |  Decode |
> > | :----: | :-----: | :-----: |
> > |    1.0 |  0.1618 | -0.1777 |
> > |    2.0 |  0.2987 | -0.3151 |
> > |    3.0 |  0.1898 | -0.3379 |
> >
> > > **Table R1:** Correlation between relative offloading overhead and average LRU hit rate measured on the full corpus, under various cache size ratio $\rho$.
> >
> > Measuring and throughputs on existing MoE models, on the other hand, yields results that may not be comparable due to the vast difference between model sizes and architectures, thus we will not report these results.
> >
> > ## A2: SCH versus LRU hit rate
> >
> > > **W2:** What advantage does SCH have compared to common cache algorithm hit rate? Can we not determine required cache size by analyzing LRU hit rate vs $\rho$?
> >
> > As introduced in A1, SCH estimates the upper bound of the hit rate of any cache algorithms applied to expert offloading systems. The advantages of SCH can be summarized as two-fold:
> >
> > 1. While LRU has been proven to be effective in many cases, it might ***not represent other algorithms such as LFU*** which we observed performed better on certain MoE models.
> > 2. LRU hit rate does not provide ***fine-grained information about the length of local routing consistency.***
> >
> > Nevertheless, given the high positive correlation between SCH, LRU and LFU hit rates, we agree that LRU hit rate may be used as an alternative when solely deciding the cache size.
> >
> > ## A3: Model architecture
> >
> > > **W3 & Q2:** In analysis in Section 3.3, the authors suggest that applying MoE on every layer and not sharing experts results in higher routing consistency. While there might be a correlation between the two, it does not imply causation. Particularly in the case of 'MoE on every layer', there are just 4 out of 20 models that apply at every $k$ layer instead and of those four, two of them have extremely high ratio of active to total experts ($1:64$ and $1:128$). Clearly explain why the dense modules might decrease local consistency (SRP).
> >
> > We agree that the observational results do not decouple the two mentioned factors enough for a robust conclusion. In order to further examine our conjectures, we conduct experiments on standard toy models modified from OLMoE that have comparable sizes (all 1.43B) and controlled MoE designs. **Table R2** illustrates parts of the results when segment length $m=16$, where we find that adding dense *layers* actually increases SRP a little, but with the cost of a higher perplexity. However, adding another dense component, *shared experts*, significantly decreases SRP and does not lower perplexity. We propose two hypothesis for the effects of shared experts: bypass effect, where shared experts process more information than the original group of experts; shrinked expert combination space, where routers have less expert combination choices for different inputs due to decreased number of total and activated experts. We will include the full results and an updated Section 3.3 in the revised manuscript.
> >
> > | Model    | SRP   | log PPL |
> > | :------: | :---: | :-----: |
> > | Baseline | 43.56 |    2.40 |
> > | 1ShrExp  | 41.38 |    2.40 |
> > | 2ShrExp  | 38.78 |    2.41 |
> > | DenseFst | 44.87 |    2.43 |
> > | DenseHlf | 43.67 |    2.46 |
> >
> > > **Table R2:** SRP ($m=16$) and log perplexity on the full corpus of several standard toy models. 1ShrExp and 2ShrExp replace 1 and 2 experts to shared experts respectively; DenseFst replaces the first layer with a dense MLP; DenseHlf does so every two layers.

---

> > > ### Author Response · Authors · 2025-12-03
> > >
> > > ## A4: Writing and presentation
> > >
> > > > **W4.i:** The proposed metrics in Section 2 require a lot of time to understand. Providing intuitive/textual explanations for the equations, removing repeated mentioning of variables (for e.g., saying expert $e\in E$ at all locations instead of just saying expert) and moving unnecessary equations to appendix and replacing them with text in the main section would vastly improve the reader’s experience.
> > >
> > > Thank you for your suggestion about the writing style in Section 2. We will provide a clearer and more intuitive version of this section and move some equations to the Appendix in the revised manuscript.
> > >
> > > > **W4.ii:** The term $\rho$ is overloaded - it is used in both SRP for segment routing size ratio and in SCH for segment cache size ratio - causing a lot of confusion.
> > >
> > > Currently we use the same notation for both concepts since they have the same core: **the fraction of selected experts to represent all expert activation within a segment**. We will use a clearer notation to distinguish the two usages when necessary in the revised manuscript.
> > >
> > > > **W4.iii & Q3:** Some of the figures are hard to parse. Is it useful to have the marker size representing model size in Figure 2? Similarly, can expert specialization vs SRP be used in the plot in Figure 4 instead of correlation? Why do LL2 and Y2 have a very low correlation while they have high SRP and relatively high expert domain specialization? Why do STe have high correlation but low domain specialization?
> > >
> > > 1. In Figure 2, the different marker sizes aim to demonstrate that SRP is not correlated to the model size; we will note this in the text and illustrate it in a more straightforward way in the revised manuscript.
> > >
> > > 2. The y-axis in Figure 4 is the correlation between expert specialization and SRP **across all experts in a model**, which reflects the relation between SRP and specialization on expert level. For example, LL2 and Y2 have experts that are always activated, which contribute a lot to SRP but are definitely not specialized. As for STe, the correlation coefficient is around than $0.3$, which is not very high compared to many other models; meanwhile its domain specialization is mediocre.
> > >
> > > ## A5: Blackbox models
> > >
> > > > **W5:** Since the proposed metric requires access to expert activation, it is not possible to measure them for black box models.
> > >
> > > We acknowledge your concern about blackbox models. However, because black box models are neither modifiable nor servable with third-party expert offloading systems, we argue that any attempt to measure SRP, SCH or other similar metrics for such models (even if succeeded) can hardly make any contribution without opening the black box.

---

### Author Response · Authors · 2025-12-03
**Rebuttal summary**

We are grateful to all reviewers for their valuable feedback and recognition of the significance of our work, including providing a novel perspective (local routing consistency) and corresponding metrics (SRP and SCH) that focus on the effect of model design on expert offloading efficiency (`jCkR`, `9czm`, `3aLj`), throrough empirical experiments across multiple MoE models and datasets (`jCkR`, `9czm`, `3aLj`), and important insights concerning model architecture, load balance, expert specialization and cache size selection (`jCkR`).

In response to the reviewers' comments and questions, we provide rebuttal arguments including (but not limited to):

- Empirical experiments on model throughput (`jCkR`, `3aLj`);
- Advantages of SCH compared to LRU hit rate (`jCkR`);
- Experiments verifying model architecture insights (`jCkR`, `9czm`);
- Writing and presentation improvements (`jCkR`, `3aLj`);
- Application to blackbox models (`jCkR`);
- Empirical analysis between LRC and LM performance (`9czm`);
- Discussion on prefilling stage performance (`9czm`);
- Further analysis of expert specialization (`3aLj`).

We hope these responses address the key concerns of the reviewers.

---

### Author Response · Authors · 2025-12-04
**Submission update**

Following our rebuttals, we have updated our submission material according to the reviewers' feedback and suggessions, including the paper and supplementary materials. Main update points include:

- We rearrange Sections 2.2 and 2.3 that introduces SRP and SCH into a more straighforward version with visualization; moving formal definitions and formulas to Appendix C.1;
- We update the definition of SCH as mentioned in the rebuttal comments;
- We add information about the toy models we have trained to validate our conclusions in Section 3.1 and Appendix D.2, and illustrate corresponding overall results in Figure 5;
- We merge the two paragraphs in Section 3.3 and update the analysis and conclusions to incorporate toy model results and demonstrate factors that affect local routing consistency according to observed significance.
- We enlarge several figures in the paper to improve their readability.
- We attach a brief discussion about the relation between local routing consistency and the throughput of real expert offloading systems in Appendix F.

We have highlighted the updated parts in the paper using red font color, and we hope that the rebuttals and updates together address the key concerns of the reviewers.

---

### Meta-Review · Area_Chair_k4cB · 2026-01-06

**Summary:**

This paper introduces local routing consistency as a key property for expert offloading in MoE LLMs, proposes two metrics (SRP and SCH), and evaluates them across 20 MoE LLMs to derive practical insights about architecture choices, specialization, load balance, and cache sizing. Reviewer sentiment is borderline (two 4’s and one 6), with the main decision-critical concerns being: (i) lack of a clear connection from SRP/SCH to real throughput/offloading performance, (ii) correlation-vs-causation in architectural claims, and (iii) clarity/presentation of the metric definitions and figures.

**Reviewer Concerns:**

Addressed by rebuttal / revisions:

+Throughput linkage: Authors provide a conceptual throughput analysis relating SCH to cache misses and overhead; they also add toy-model throughput/offloading experiments (LRU-based) and discuss decoding vs prefilling behavior.

+SCH vs standard cache hit-rate (e.g., LRU): Authors argue SCH upper-bounds achievable hit rates and provides finer-grained locality characterization; they also note strong correlation with LRU/LFU and accept LRU as a practical alternative for cache sizing.

+Correlation vs causation for architecture insights: Authors add controlled toy-model ablations (dense layers vs shared experts) and update conclusions accordingly (dense layers may slightly increase SRP but with worse perplexity; shared experts reduce SRP).

+Presentation: Authors commit to restructuring Section 2 (more visual/intuitive, definitions moved to appendix), enlarging figures, and fixing overloaded notation.

+Prefill vs decode: Authors explicitly discuss prefilling and argue load balance dominates there, while routing consistency matters more for decoding.

Still outstanding (but not necessarily blocking):

- No end-to-end validation on a real offloading system for real MoE LLMs; evidence is primarily proxy metrics + toy-model throughput correlations, so claims about “guiding deployment” should be stated with appropriate caveats.

- Some insights remain observational even after toy-model support (useful, but should be framed as evidence-backed guidance rather than definitive causal rules).

**Reviewer Scores:**

If full discussion had occurred, I expect modest upward movement due to concrete rebuttal additions and planned clarity fixes:

Reviewer jCkR: 4 → 5 (main asks were throughput connection, SCH vs LRU, causality/clarity; rebuttal addresses all with analysis + toy experiments + planned rewrite).

Reviewer 9czm: 4 → 5 (key concerns were causality, perf trade-off discussion, prefilling, and “actionability”; rebuttal adds toy ablations, discusses LM perf vs SRP, and separates prefill vs decode).

Reviewer 3aLj: 6 → 6 (maybe 7) (most concerns were presentation and lack of end-to-end offloading eval; rebuttal adds toy throughput discussion and significant presentation revisions, but real-system evidence remains limited).

---

### Decision · Program_Chairs · 2026-01-26

Accept (Poster)